# Foundation VAEs for 3D CT Reconstruction, Augmentation, and Generation

Qi Chen [* 1]   Shuhan Ding [* 2]   Yu Gu [3]   Nan Liu [2]   Jiang Bian [3]   Alan Yuille [1]   Zongwei Zhou [1]   Jingjing Fu [3]

## Abstract

Variational autoencoders (VAEs) compress high resolution CT volumes into compact latents while preserving clinically relevant structure. However, training CT-specific VAEs from scratch or heavily fine-tuning them incurs substantial computational and engineering cost, and often degrades under heterogeneous scanners, protocols, and diseases. This paper makes a progressive stride toward training-free medical VAEs by leveraging a critical observation: a single Foundation VAE, pretrained at scale on natural images and videos, can serve as a unified interface for CT Reconstruction, Augmentation, and Generation. With both encoder and decoder frozen, the Foundation VAE reconstructs CT volumes with preserved anatomy while suppressing acquisition noise; training segmentation models on these reconstructions improves surface accuracy by 3.9% NSD on average for pancreatic tumor and lung tumor. Within the same Foundation VAE latent space, a conditional latent diffusion model achieves 3.9% lower average FVD with 36.2% higher CT CLIP score, and improves multi-disease generation faithfulness across 18 types by 2.76% AUC. These results demonstrate Foundation VAEs as a practical interface for scalable CT representation reuse and faithful CT generation. Our code and demo are available at https://github.com/qic999/Foundation-VAE.

## 1. Introduction

Variational autoencoders (VAEs) (Kingma & Welling, 2013) have become a core representation interface for modern reconstruction and generative modeling by compressing high

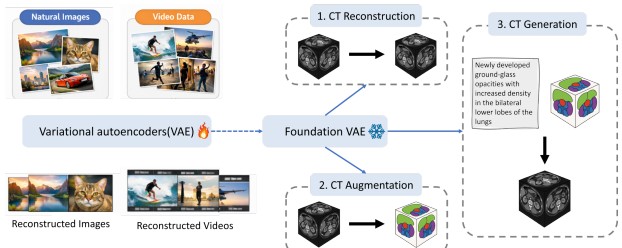

*Figure 1.* **Foundation VAE for CT Reconstruction, CT Augmentation, and CT Generation.** (1) *CT Reconstruction:* A *Foundation VAE* pretrained at scale on natural images and videos reconstructs a 3D CT volume via its frozen encoder $E$ and decoder $D$. (2) *CT Augmentation:* Through zero shot transfer to CT, the reconstructed volumes provide a boundary enhanced training view, improving downstream segmentation especially on surface accuracy. (3) *CT Generation:* In the fixed latent space of the same *Foundation VAE*, we train a conditional latent diffusion model to synthesize anatomically consistent healthy and abnormal CT volumes, controlled by organ masks and clinical findings.

dimensional signals into compact latents while preserving essential structure. This interface is particularly attractive for three dimensional computed tomography (CT): volumes are acquired at high resolution with large fields of view, making pixel space learning expensive, while latent space modeling can substantially reduce computation and memory demands without discarding clinically relevant cues.

Despite rapid progress, most CT reconstruction and CT generation pipelines still depend on a costly domain specific representation stage. In practice, systems train a CT tailored encoder and decoder (often a VAE) from scratch or heavily fine tune it on large medical corpora, then learn diffusion models or other generators on the resulting latents. This design increases compute cost and engineering complexity, and it can degrade under heterogeneous scanners, acquisition protocols, and diverse disease patterns. Tables 1 and 2 illustrate the generalization risk: MedVAE (Varma et al., 2025a) shows a clear reconstruction collapse on MSD dataset (Antonelli et al., 2022), with PSNR dropping to 20.34 and SSIM to 0.52 on Lung, and PSNR 18.78 with SSIM 0.33 on Pancreas, alongside MSE exceeding 600 and 1000, suggesting overfitting to its training distribution. MAISI (Guo et al., 2025b) yields stronger reconstruction,

[1]Department of Computer Science, Johns Hopkins University [2]Duke-NUS Medical School [3]Microsoft Research. Correspondence to: Jingjing Fu <jifu@microsoft.com>.

*Proceedings of the $43^{rd}$ International Conference on Machine Learning*, Seoul, South Korea. PMLR 306, 2026. Copyright 2026 by the author(s).

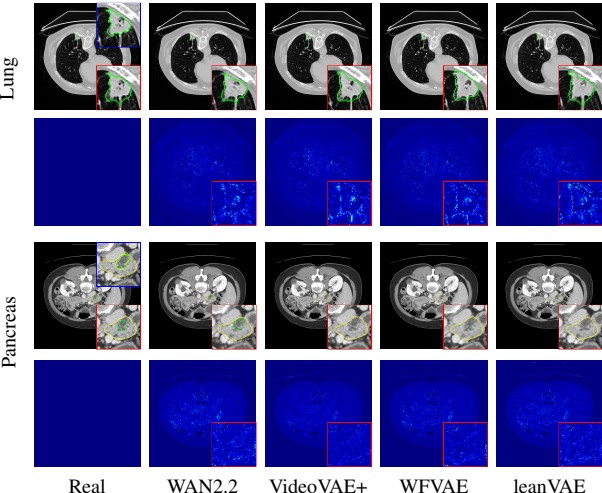

*Figure 2.* **CT reconstruction and segmentation comparison** of off-the-shelf video VAEs without medical fine-tuning on MSD (Antonelli et al., 2022) Task06 Lung and Task07 Pancreas. Top row shows reconstructions with segmentation overlays, bottom row shows voxel-wise error maps. Red insets zoom in lesion regions and the blue inset in the Real column shows ground-truth labels. Green contours denote tumors and yellow contours denote organs, with missing contours indicating failed segmentation. Errors are dominated by high-frequency noise and mild streak artifacts rather than boundary shifts, consistent with $T(\cdot)$ acting as a boundary-preserving denoiser.

but it requires large scale 3D VAE training, using 37,243 CT volumes, 8 32G V100 GPUs, and 300 epochs with multi stage patch cropping from $[64, 64, 64]$ to $[128, 128, 128]$. When the representation model is mismatched to new data, downstream models inherit its limitations, leading to distorted anatomy, inconsistent pathology appearance, and reduced usefulness for clinical tasks.

We revisit a transfer hypothesis: a *Foundation VAE* pretrained at scale on natural images and videos can be transferred as a general CT interface without medical fine-tuning. The central finding is that a single *Foundation VAE* supports *CT Reconstruction*, *CT Augmentation*, and *CT Generation* within one shared latent space, avoiding a separate CT-specific representation stage.

The reconstruction discrepancy concentrates on high frequency grain and mild scanner dependent artifacts, while organ and lesion boundaries remain spatially aligned (Figure 2). The error maps and zoomed in insets indicate noise attenuation rather than boundary shifts, consistent with the frozen encoder and decoder behaving as a boundary stable reconstruction operator for CT (Figure 2). Quantitatively, across MSD Lung and Pancreas, off the shelf *Foundation VAE* reconstructions achieve strong PSNR and SSIM with low MSE (Table 1 and 2). In contrast, MedVAE collapses on MSD with markedly worse PSNR and SSIM and much higher MSE, suggesting limited generalization beyond its

training distribution.

Because boundary geometry is preserved, reconstructed CT volumes remain task useful for segmentation. Training segmentation on reconstructed volumes matches or improves performance compared to training on real CT, with the most pronounced gains on NSD, a surface based metric that directly reflects boundary quality. This aligns with cleaner transitions around organ and lesion contours that yield less ambiguous boundary neighborhoods. Using reconstructed volumes as an additional training view yields an average gain of 3.9% NSD on pancreatic tumor and lung tumor.

Beyond reconstruction, the same fixed *Foundation VAE* latent space provides a compact and stable feature interface for *CT Generation*. A conditional latent diffusion model trained in this latent space is grounded by anatomy masks as spatial constraints and radiology reports as semantic constraints, and is further strengthened by a lightweight three dimensional consistency module that encourages coherent anatomy and pathology across axial slices. The resulting model achieves 3.9% lower average FVD with 36.2% higher CT CLIP score, and improves multi disease *CT Generation* faithfulness across 18 types by 2.76% AUC.

Our contributions are summarized as follows:

- **Foundation VAE as a CT interface.** We show that a VAE pretrained on natural images and videos can serve as a unified representation interface for CT Reconstruction, CT Augmentation, and CT Generation without medical fine tuning.

- **Reconstruction based augmentation.** We demonstrate that training with frozen VAE reconstructions improves boundary accuracy for downstream segmentation, with consistent gains on pancreas and lung tumor benchmarks.

- **Conditional latent diffusion for CT Generation.** We train diffusion models in the fixed Foundation VAE latent space with anatomy and radiology report conditioning and a three dimensional consistency module, enabling controllable, multi-disease CT Generation within a single unified generator.

**Conflict of Interest Disclosure.** The authors declare that they have no competing financial interests or personal relationships that could have influenced the work reported in this paper.

## 2. CT Reconstruction & Augmentation

We observe that a *Foundation VAE*, pretrained at scale on natural images and videos, can be transferred to 3D CT as a zero shot pixel level interface (Huix et al., 2024; Noh & Lee,

*Table 1.* Reconstruction performance across four CT datasets. Details of the VAEs are provided in Appendix B.

| Model | Task06 Lung | | | Task07 Pancreas | | | LiTS | | KiTS19 | |
|---|---|---|---|---|---|---|---|---|---|---|
| | PSNR↑ | SSIM↑ | MSE↓ | PSNR↑ | SSIM↑ | MSE↓ | PSNR↑ | SSIM↑ | PSNR↑ | SSIM↑ |
| WAN2.1 | $30.93 \pm 4.06$ | $0.76 \pm 0.10$ | $77.97 \pm 55.12$ | $39.18 \pm 1.38$ | $0.94 \pm 0.02$ | $8.58 \pm 2.93$ | $39.32 \pm 1.69$ | $0.93 \pm 0.03$ | $40.22 \pm 1.68$ | $0.94 \pm 0.03$ |
| WAN2.2 | $30.93 \pm 4.06$ | $0.76 \pm 0.10$ | $77.97 \pm 55.12$ | $39.06 \pm 1.50$ | $0.95 \pm 0.02$ | $8.99 \pm 3.31$ | $39.25 \pm 1.84$ | $0.94 \pm 0.03$ | $40.32 \pm 1.84$ | $0.95 \pm 0.03$ |
| VideoVAE+ | $30.94 \pm 4.35$ | $0.77 \pm 0.11$ | $80.43 \pm 59.08$ | $40.12 \pm 1.66$ | $0.95 \pm 0.02$ | $7.00 \pm 3.05$ | $39.92 \pm 1.97$ | $0.95 \pm 0.03$ | $41.07 \pm 1.91$ | $0.96 \pm 0.03$ |
| IVVAE | $31.78 \pm 4.11$ | $0.79 \pm 0.10$ | $64.39 \pm 45.95$ | $40.43 \pm 1.54$ | $0.96 \pm 0.02$ | $6.45 \pm 2.52$ | $40.33 \pm 1.89$ | $0.95 \pm 0.02$ | $41.38 \pm 1.90$ | $0.96 \pm 0.03$ |
| CVVAE | $29.61 \pm 3.27$ | $0.75 \pm 0.10$ | $93.91 \pm 57.96$ | $35.34 \pm 0.90$ | $0.93 \pm 0.02$ | $20.87 \pm 4.14$ | $36.46 \pm 1.19$ | $0.93 \pm 0.03$ | $36.63 \pm 1.61$ | $0.93 \pm 0.03$ |
| WFVAE | $30.98 \pm 4.29$ | $0.78 \pm 0.10$ | $79.23 \pm 58.11$ | $39.53 \pm 1.43$ | $0.95 \pm 0.02$ | $7.99 \pm 2.83$ | $40.04 \pm 1.79$ | $0.95 \pm 0.02$ | $40.79 \pm 1.81$ | $0.96 \pm 0.03$ |
| LeanVAE | $30.66 \pm 4.33$ | $0.78 \pm 0.10$ | $86.29 \pm 62.76$ | $39.29 \pm 1.44$ | $0.95 \pm 0.02$ | $8.50 \pm 3.10$ | $39.64 \pm 1.78$ | $0.95 \pm 0.03$ | $40.53 \pm 1.80$ | $0.95 \pm 0.03$ |
| MedVAE | $30.06 \pm 3.60$ | $0.74 \pm 0.11$ | $88.36 \pm 59.05$ | $36.00 \pm 1.13$ | $0.91 \pm 0.03$ | $17.98 \pm 5.78$ | $34.11 \pm 5.13$ | $0.85 \pm 0.11$ | $33.76 \pm 7.17$ | $0.91 \pm 0.06$ |
| MAISI | $29.78 \pm 2.99$ | $0.73 \pm 0.09$ | $86.25 \pm 54.54$ | $36.97 \pm 0.92$ | $0.93 \pm 0.02$ | $13.88 \pm 3.09$ | $37.35 \pm 1.23$ | $0.92 \pm 0.02$ | $37.08 \pm 1.30$ | $0.92 \pm 0.03$ |

*Table 2.* Segmentation performance of nnU-Net trained on data reconstructed by each VAE across four CT datasets. For Task07 Pancreas, LiTS, and KiTS19, indices 1 and 2 denote organ and tumor/lesion classes respectively.

| Model | Task06 Lung | | Task07 Pancreas | | | | LiTS | | KiTS19 | |
|---|---|---|---|---|---|---|---|---|---|---|
| | DSC↑ | NSD↑ | DSC1↑ | NSD1↑ | DSC2↑ | NSD2↑ | DSC1↑ | DSC2↑ | DSC1↑ | DSC2↑ |
| Real Data | $66.4 \pm 29.2$ | $70.2 \pm 31.6$ | $82.2 \pm 8.0$ | $79.2 \pm 9.6$ | $42.8 \pm 32.8$ | $39.8 \pm 32.4$ | $94.7 \pm 5.3$ | $57.2 \pm 27.8$ | $95.5 \pm 3.9$ | $83.2 \pm 19.1$ |
| WAN2.1 | $71.9 \pm 24.7$ | $75.3 \pm 28.7$ | $82.2 \pm 8.0$ | $79.0 \pm 10.2$ | $45.2 \pm 31.9$ | $41.9 \pm 31.9$ | $94.3 \pm 6.7$ | $57.1 \pm 29.3$ | $95.4 \pm 4.4$ | $83.6 \pm 18.2$ |
| WAN2.2 | $70.2 \pm 20.9$ | $72.9 \pm 24.2$ | $82.0 \pm 8.1$ | $79.0 \pm 9.5$ | $47.2 \pm 31.7$ | $45.0 \pm 31.8$ | $95.1 \pm 4.9$ | $60.8 \pm 26.8$ | $96.0 \pm 3.5$ | $85.0 \pm 15.6$ |
| VideoVAE+ | $68.0 \pm 21.3$ | $72.6 \pm 24.2$ | $82.2 \pm 8.3$ | $79.3 \pm 10.0$ | $47.1 \pm 32.7$ | $44.7 \pm 32.2$ | $94.8 \pm 5.1$ | $58.2 \pm 26.9$ | $95.7 \pm 3.6$ | $83.6 \pm 18.7$ |
| IVVAE | $70.2 \pm 20.5$ | $73.3 \pm 24.0$ | $82.0 \pm 8.3$ | $78.5 \pm 10.4$ | $47.2 \pm 31.8$ | $44.3 \pm 32.3$ | $95.3 \pm 4.7$ | $61.6 \pm 26.4$ | $95.7 \pm 3.5$ | $83.8 \pm 18.7$ |
| CVVAE | $67.0 \pm 26.7$ | $70.9 \pm 29.2$ | $79.4 \pm 9.2$ | $74.7 \pm 10.4$ | $36.7 \pm 31.3$ | $34.5 \pm 28.8$ | $93.5 \pm 8.1$ | $52.4 \pm 29.6$ | $94.5 \pm 4.0$ | $78.8 \pm 23.3$ |
| WFVAE | $68.0 \pm 27.4$ | $71.3 \pm 29.0$ | $81.3 \pm 8.4$ | $77.9 \pm 10.8$ | $46.9 \pm 30.2$ | $44.4 \pm 30.2$ | $95.1 \pm 4.4$ | $59.5 \pm 27.0$ | $95.4 \pm 4.3$ | $85.5 \pm 14.7$ |
| LeanVAE | $69.2 \pm 21.6$ | $72.5 \pm 24.7$ | $82.0 \pm 8.0$ | $78.6 \pm 10.1$ | $44.1 \pm 32.8$ | $41.9 \pm 31.7$ | $94.9 \pm 5.0$ | $59.4 \pm 26.5$ | $95.5 \pm 4.4$ | $83.7 \pm 16.3$ |
| MedVAE | $71.5 \pm 19.3$ | $74.7 \pm 23.6$ | $80.7 \pm 8.7$ | $77.0 \pm 10.7$ | $42.4 \pm 33.6$ | $40.5 \pm 33.4$ | $94.3 \pm 6.0$ | $57.4 \pm 28.0$ | $94.8 \pm 3.5$ | $77.9 \pm 25.4$ |
| MAISI | $71.0 \pm 19.5$ | $74.5 \pm 21.0$ | $80.3 \pm 8.7$ | $75.8 \pm 10.5$ | $35.0 \pm 31.9$ | $32.0 \pm 30.2$ | $93.6 \pm 7.7$ | $38.5 \pm 34.5$ | $94.4 \pm 5.2$ | $79.5 \pm 22.5$ |

2025). Given a CT volume $x$, we apply the frozen encoder $E$ and decoder $D$ to obtain

$$\tilde{x} := T(x) = D(E(x)). \tag{1}$$

Although $E$ and $D$ are never exposed to medical data, $T(\cdot)$ produces reconstructions that preserve clinically relevant geometry and remain directly useful for downstream segmentation. This observation suggests that a large scale VAE prior can act as a CT compatible reconstruction operator that suppresses nuisance variability while preserving task relevant boundaries (Venkatakrishnan et al., 2013; Vincent et al., 2008). In light of this, we validate the pixel level role of $T(\cdot)$ through two empirical analyses and a theoretical justification.

**(1) CT Reconstruction: the reconstruction gap is dominated by CT noise, not structural distortion.** We observe that the discrepancy between a real CT volume $x$ and its reconstruction $\tilde{x}$ follows the statistics of CT acquisition and reconstruction noise, rather than anatomy mismatch. As visualized in Figure 4, voxel wise difference maps concentrate on grain like fluctuations in low contrast soft tissue regions and mild scanner dependent artifacts, while organ and lesion boundaries remain spatially aligned. This indicates that the VAE bottleneck primarily attenuates weakly structured high frequency components instead of shifting anatomical surfaces. Quantitatively, Tables 1 and 2 shows that off the

shelf video VAEs achieve reconstruction quality comparable to a CT specific VAE, and the deviation between $x$ and $\tilde{x}$ is consistently summarized by voxel wise MSE. Together, these results support interpreting $T(\cdot)$ as a boundary stable CT reconstruction operator.

**(2) CT Augmentation: boundary preservation explains stable segmentation, and training on reconstructions improves robustness.** Segmenters trained on reconstructed volumes $\tilde{x}$ achieve performance comparable to, and often better than, training on the original scans $x$, as shown in Table 1. The improvement is most pronounced on NSD, a surface based metric that directly reflects boundary quality, where Table 1 reports consistent and sometimes large gains across organs and tumors. This trend aligns with the qualitative evidence in Figure 2, where reconstructions exhibit cleaner transitions near organ and lesion contours, yielding sharper local gradients and less ambiguous boundary neighborhoods. Based on this, our CT augmentation is simple: for each labeled pair $(x, y)$, we train the segmenter directly on $(\tilde{x}, y)$, using the reconstruction as the primary training view. This is annotation free since labels are unchanged, and it consistently improves cross dataset robustness, especially on boundary sensitive metrics.

**(3) Theoretical justification: segmentation risk is preserved under task relevant reconstruction stability.** We provide a formal justification that connects boundary stabil-

ity of $T(\cdot)$ to the observed segmentation gains. Specifically, we show that if $T(\cdot)$ satisfies a task relevant stability condition that keeps label defining geometry approximately invariant, then training on reconstructed inputs induces a bounded segmentation risk gap compared to training on the original inputs. The proof follows a standard decomposition of excess risk into a reconstruction induced perturbation term and a task Lipschitz term, yielding an explicit upper bound that scales with the stability radius of $T(\cdot)$. Detailed assumptions, theorem statement, and step by step proof are provided in Appendix A.

## 3. CT Generation

We observe that the latent space of a *Foundation VAE* provides a compact and stable representation for *CT Generation*. Leveraging this property, we use *Foundation VAE* features to support both *healthy CT generation* and *abnormal CT generation* under a unified latent synthesis framework. We keep the *Foundation VAE* encoder $\mathcal{E}$ and decoder $\mathcal{D}$ frozen, and train a conditional latent diffusion model in this fixed latent space. CT volumes are generated under anatomical and clinical controls, as shown in Figure 3.

Given a CT volume $\mathbf{x}$, we obtain its latent code

$$\mathbf{z}_0 = \mathcal{E}(\mathbf{x}). \tag{2}$$

We define the forward diffusion process in latent space as

$$\begin{aligned} \mathbf{z}_t &= \sqrt{\bar{\alpha}_t}\,\mathbf{z}_0 + \sqrt{1 - \bar{\alpha}_t}\,\boldsymbol{\epsilon}, \\ \boldsymbol{\epsilon} &\sim \mathcal{N}(\mathbf{0}, \mathbf{I}), \qquad t = 1, \dots, T, \end{aligned} \tag{3}$$

and train a denoising U Net $\epsilon_\theta$ conditioned on anatomy $\mathbf{m}$ and radiology reports $r$ to predict $\boldsymbol{\epsilon}$:

$$\mathcal{L} = \mathbb{E}_{\mathbf{x}, \boldsymbol{\epsilon}, t, \mathbf{m}, r}\left[\left\|\boldsymbol{\epsilon} - \epsilon_\theta([\mathbf{z}_t; \mathbf{z}_m], r, t)\right\|_2^2\right]. \tag{4}$$

### 3.1. Conditioning on Anatomy and Radiology Reports

We support two synthesis modes under a unified conditioning interface. For *healthy CT generation*, we condition on an organ mask $\mathbf{m}^{\mathrm{org}}$ that specifies body anatomy, together with a normal radiology report description $r_{\mathrm{healthy}}$. For *abnormal CT generation*, we additionally condition on a disease mask $\mathbf{m}^{\mathrm{dis}}$ and a radiology report description $r_{\mathrm{abnormal}}$ that specifies pathology attributes.

**Mask anatomy embedding.** We obtain $\mathbf{m}^{\mathrm{org}}$ using a pretrained organ segmentation model (He et al., 2025; Wasserthal et al., 2023), which provides masks for 124 anatomical structures (see Appendix C for details). We form the mask volume

$$\mathbf{m} = \left[\mathbf{m}^{\mathrm{org}}, \mathbf{m}^{\mathrm{dis}}\right], \tag{5}$$

where for healthy cases we set $\mathbf{m}^{\mathrm{dis}} = \mathbf{0}$, and for abnormal cases we provide a nonzero $\mathbf{m}^{\mathrm{dis}}$. To align the spatial condition with latent diffusion, we encode the mask volume with the same frozen *Foundation VAE* encoder $\mathcal{E}(\cdot)$ to obtain a mask embedding

$$\mathbf{z}_m = \mathcal{E}(\mathbf{m}). \tag{6}$$

During denoising, we concatenate $\mathbf{z}_m$ with the noised latent $\mathbf{z}_t$ along the channel dimension and feed the concatenated tensor into the denoising U-Net,

$$\hat{\boldsymbol{\epsilon}} = \epsilon_\theta\big([\mathbf{z}_t; \mathbf{z}_m], r, t\big), \tag{7}$$

so the generator is explicitly grounded to organ and lesion geometry throughout the diffusion process.

**Text report injection.** We encode the text condition $r \in \{r_{\mathrm{healthy}}, r_{\mathrm{abnormal}}\}$ using a frozen text encoder $\tau(\cdot)$ to obtain text embeddings. As shown in Figure 3, we inject these embeddings into the denoising U-Net via cross attention, enabling the model to align synthesized textures and patterns with clinical language.

### 3.2. 3D consistency

To ensure consistent texture within each axial slice and coherent structures across slices, we append a lightweight 3D consistency attention. Let $X \in \mathbb{R}^{B \times C \times F \times H \times W}$ denote $F$ consecutive axial slices in latent space. Each slice index $f$ is updated by aggregating information from neighboring slices along the through-slice axis:

$$Y_{b,c,f,h,w} = \sum_{\tau = -\lfloor k_s/2 \rfloor}^{\lfloor k_s/2 \rfloor} w_{c,\tau} X_{b,c,f+\tau,h,w}, \tag{8}$$

where $k_s$ is the slice-axis kernel size and $w_{c,\tau}$ are learnable weights shared across $(h, w)$. The kernel is initialized as a Dirac delta (identity mapping), allowing the model to gradually learn smooth axial textures and coherent volumetric structures that match physiological continuity across slices.

## 4. Evaluation of CT Generation

### 4.1. Experimental Setup

**Datasets.** CT-RATE (Ai et al., 2024) and ReXGroundingCT (Baharoon et al., 2025) are used as the primary datasets. Overall, the dataset contains 4,961 training volumes and 80 test volumes, spanning normal cases and 18 disease categories. Specifically, a *normal subset* is constructed from CT-RATE with 2,395 no-disease training volumes and 30 no-disease test volumes. A *diseased subset* is defined as the overlap between CT-RATE and ReXGroundingCT, resulting in 2,566 diseased training volumes with 6,342 disease masks and a diseased test set of 50 volumes with 297

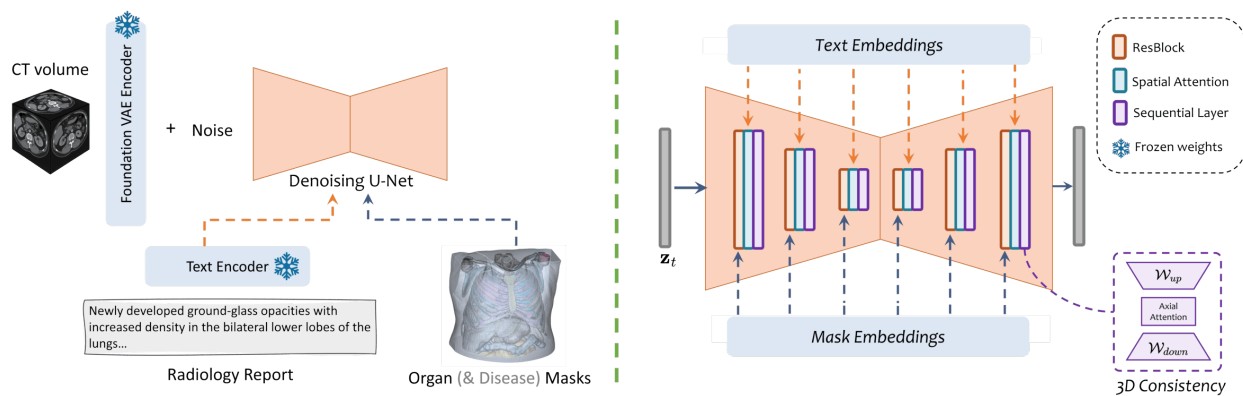

*Figure 3.* **CT Generation with Foundation VAE.** Building on *Foundation VAE*, healthy and abnormal CT volumes are generated in the fixed latent space of the frozen *Foundation VAE*. The architecture consists of two parts: (1) *Conditioning on Anatomy and Radiology Reports* (§ 3.1), where organ and disease masks are encoded by the same frozen *Foundation VAE* and concatenated with the noised latent at each denoising block for spatial grounding, while radiology report embeddings from a frozen text encoder are injected via cross attention; (2) *3D consistency* (§ 3.2), a lightweight attention that encourages coherent anatomy and pathology across axial slices.

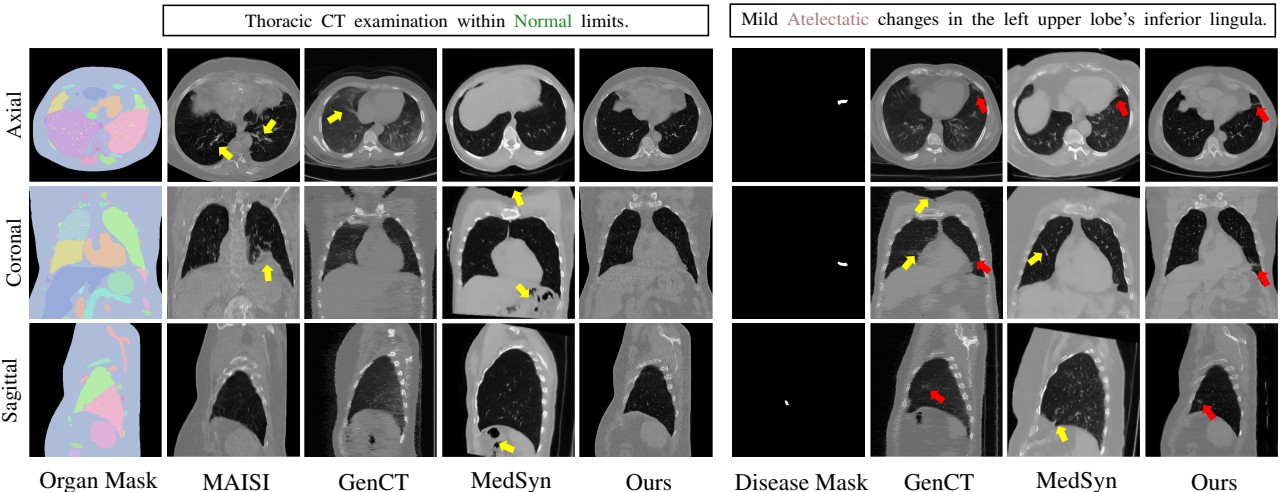

*Figure 4.* Axial, sagittal, and coronal slices of 3D CT volumes synthesized by different methods under the same prompts. Our method generates spatially consistent and anatomically detailed volumes that better align with organ structure and disease-mask guidance. Red arrows indicate correctly synthesized atelectasis, while yellow arrows mark incorrect or spurious findings.

disease masks. For the downstream multi-label classification task, 500 training volumes and 200 test volumes are additionally sampled from CT-RATE. Each CT volume is paired with its radiology report from CT-RATE. Disease masks are provided by ReXGroundingCT. Organ masks are obtained by combining VISTA3D (He et al., 2025) and To-talSegmentator (Wasserthal et al., 2023) to form whole-body anatomical segmentation. All volumes are resampled to $512 \times 512$ per slice (100–600 slices per volume), and intensities are clipped to $[-1000, 1000]$ HU. Additional dataset and implementation details are provided in Appendix. C.

**Evaluation Metrics.** The quality of generated CT volumes is comprehensively evaluated using Fréchet Video Distance (FVD), Fréchet Inception Distance (FID), and

CT-CLIP, measuring volumetric coherence, fidelity, and semantic consistency. Volumetric realism and slice-to-slice consistency are measured by Fréchet Video Distance (FVD), computed on features extracted by the CT-CLIP vision encoder (Hamamci et al., 2024). Lower values indicate better 3D coherence. Fidelity is measured by 2.5D Fréchet Inception Distance (FID). Each volume is sliced along the axial, sagittal, and coronal planes and encoded by a fixed RadImageNet ResNet-50 (Mei et al., 2022). Lower values indicate closer alignment to real CT features. Semantic alignment is evaluated using CT-CLIP (Hamamci et al., 2024) by cosine similarity for image-to-image (I2I) and text-to-image (T2I) retrieval. Higher scores indicate text–image consistency and greater anatomical and pathology fidelity.

*Table 3.* Quantitative comparison with state-of-the-art CT generators on the normal and disease validation subsets. Best results are in **bold**.

| Split | Method | FVD$_{\text{CT-CLIP}}$ ↓ | FID ↓ | | | | CT-CLIP ↑ | | | Inference | |
|---|---|---|---|---|---|---|---|---|---|---|---|
| | | | Axial | Sagittal | Coronal | Avg | I2I | T2I | Avg | Memory | Time/Image |
| Normal | GenerateCT | 0.5738 | 12.53 | 18.59 | 15.09 | 15.40 | 3.40 | 1.30 | 2.35 | 80G | 230s (512×512×201) |
| | MedSyn | 0.7048 | 10.37 | 13.97 | 12.16 | 12.17 | 22.99 | 27.80 | 25.40 | **7G** | 180s (256×256×256) |
| | MAISI | 0.4444 | 6.76 | 7.17 | 10.55 | 8.16 | – | – | – | 30G | 590s (512×512×128) |
| | **Ours** | **0.3035** | **2.19** | **2.32** | **2.36** | **2.29** | 76.48 | 42.23 | 59.35 | 22G | **190s (512×512×128)** |
| Disease | GenerateCT | 0.8265 | 14.50 | 26.11 | 26.19 | 22.26 | 13.26 | 6.83 | 10.05 | 80G | 230s (512×512×201) |
| | MedSyn | 0.6318 | 7.69 | 12.13 | 8.87 | 9.56 | 14.35 | 11.66 | 13.01 | **7G** | 180s (256×256×256) |
| | MAISI | 0.6433 | 4.79 | 6.11 | 8.44 | 6.45 | – | – | – | 30G | 590s (512×512×128) |
| | **Ours** | **0.5088** | **4.78** | **4.11** | **4.17** | **4.35** | 59.24 | 43.73 | 51.49 | 22G | **190s (512×512×128)** |

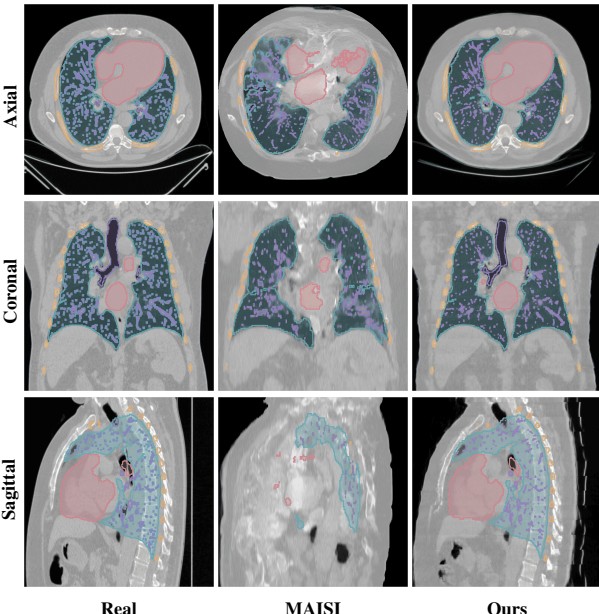

*Figure 5.* Comparison of organ grounding between Real, MAISI, and Ours using pre-trained VISTA3D and TotalSegmentator segmentations. Lung, heart, vessels, and ribs are shown as semitransparent overlays on the center axial, coronal, and sagittal slices. Our method follows the target organ masks more faithfully, producing sharper boundaries and improved alignment for thin structures.

## 4.2. Quantitative Evaluation of CT Generation.

Comparisons with text-conditioned baselines (GenerateCT (Hamamci et al., 2023), MedSyn (Xu et al., 2023)) and an anatomy-mask-conditioned baseline (MAISI (Guo et al., 2025b)) are summarized in Table 3, under both normal and disease prompts. Our method achieves strong volumetric coherence and fidelity, reaching FVD 0.30 and FID$_{\text{Avg}}$ 2.29 under no-disease prompts, and maintaining competitive coherence under disease prompts with FVD 0.51 and FID$_{\text{Avg}}$ 4.35. FID remains well balanced across axial, sagittal, and coronal views, indicating stable cross-view geometry and fewer view-dependent artifacts, whereas other generators often perform best in the axial view but degrade more noticeably in sagittal or coronal planes. Semantic alignment is also substantially improved, with CT-CLIP Avg scores of

59.35 for no-disease prompts and 51.49 for disease prompts, exceeding the baselines by a large margin. These results suggest that anatomically consistent structure supports more faithful expression of pathology-related semantics. Overall, performance decreases slightly under disease prompts, which is expected given the finer-grained variations and richer descriptions in disease synthesis.

The inference cost of our method is moderate. High-resolution 3D generation and additional spatial conditioning increase computation and memory demand, but the cost remains practical given the consistent gains in coherence, fidelity, and semantic consistency.

## 4.3. Qualitative Analysis and Cross-view Consistency

Representative three-view comparisons are shown in Figure 4. Mask-conditioned generation produces clearer structures and better slice-to-slice continuity, as seen for our method and MAISI, whereas text-only baselines often exhibit blurring or discontinuities that are most evident in the sagittal view.

The normal example shows that MAISI largely preserves mask-constrained anatomy, but without text conditioning it may fail to suppress undesired pathological patterns. Text-conditioned baselines can produce plausible textures, yet their generations may remain clinically inconsistent. GenerateCT introduces spurious abnormalities, and MedSyn can distort anatomical structures. By jointly conditioning on aligned text and organ masks, our method produces cleaner normal volumes while maintaining anatomically plausible structures.

For disease presentation, GenerateCT and MedSyn infer lesion location from text alone. Although an atelectasis-like pattern may appear in the axial view, the manifestation is often inconsistent across sagittal and coronal planes, and additional suspicious regions can emerge beyond the intended target. Conditioning on an explicit disease mask anchors abnormalities to the intended region and improves pathology–anatomy consistency across views.

## 4.4. Anatomical and Pathological Grounding Analysis

**Spatial fidelity to mask constraints.** Organ-mask adherence is quantified by applying pre-trained VISTA3D and TotalSegmentator to segment lungs, heart, pulmonary vessels, and ribs on synthesized volumes, and computing Dice/IoU against the corresponding target organ masks used for conditioning. As shown in Table 4, the proposed approach consistently improves over MAISI across all evaluated organs, with particularly large gains on thin structures. For vessels, Dice increases from 13.50 to 63.38, and for ribs from 15.20 to 70.23, indicating substantially tighter grounding beyond coarse organ shapes. Improvements are also observed for large organs, as reflected by higher lung and heart Dice scores. Figure 5 provides qualitative comparisons. Organ boundaries are sharper and better aligned with the target anatomy, and improved rib following yields a more plausible thoracic cage configuration. Vessel structures remain the most challenging, with occasional loss of fine branches and local discontinuities, suggesting further improvement is needed for high-frequency anatomical details.

**Multi-disease synthesis.** Controllable synthesis across multiple disease types is further illustrated in Figure 6 and more examples are provided in Appendix. D. In addition to location control, the generated abnormalities exhibit disease-specific appearances, with ground-glass opacity showing diffuse, hazy patterns and pleural effusion presenting as higher-density fluid-like regions. Across the z-axis, the model maintains coherent lesion morphology and stable anatomical transitions, indicating strong spatial consistency and controllability.

Pathology hallucination is a common failure mode, where generated CTs exhibit spurious abnormalities beyond the intended condition. Representative cases are shown in Figure 4. Under normal prompts, generators still introduce abnormal patterns. When the prompt specifies atelectasis in the left upper lobe, additional unintended findings may also appear elsewhere. Our method suppresses such hallucinations by combining mask constraints with aligned text supervision. Organ masks enforce anatomically valid structure, while text–mask alignment helps distinguish normal organ appearance from pathology-specific patterns under no-disease prompts. For disease synthesis, the disease mask explicitly anchors the abnormality to the target region, reducing off-target generation and improving localization consistency across views.

## 4.5. Downstream Application

To examine the utility of generative data on downstream applications, we evaluated our model in a multi-label disease classification task. A baseline classifier (Draelos et al., 2021) is trained on 500 real volumes and evaluated on 200 held-

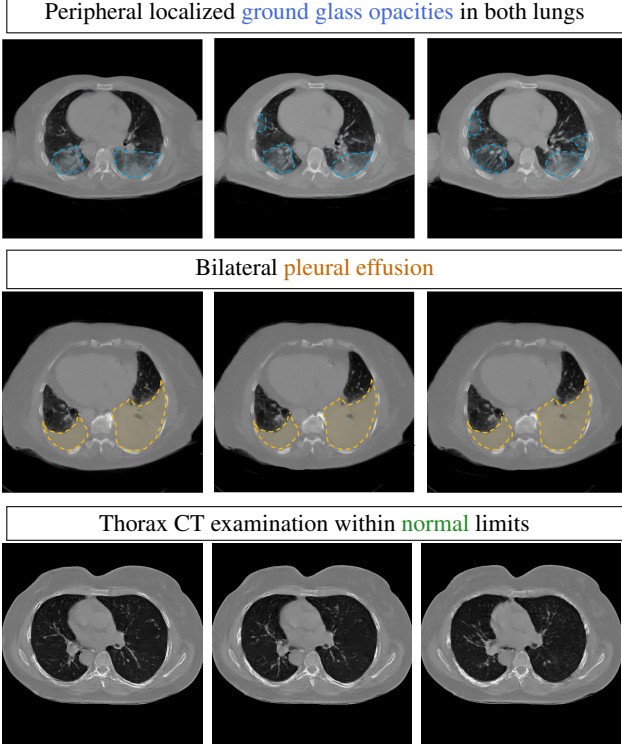

Peripheral localized ground glass opacities in both lungs

Bilateral pleural effusion

Thorax CT examination within normal limits

*Figure 6.* Controllable CT generation across multiple disease types with targeted disease-mask overlays. The synthesized abnormalities match the specified regions and exhibit disease-consistent appearances, and remaining anatomically coherent.

*Table 4.* Organ-mask adherence measured by pre-trained multi-organ segmenters on synthesized volumes. Best results are in **bold**.

| Method | Lung Dice | Lung IoU | Heart Dice | Heart IoU | Vessel Dice | Vessel IoU | Rib Dice | Rib IoU |
|---|---|---|---|---|---|---|---|---|
| MAISI | 75.94 | 62.97 | 66.86 | 52.20 | 13.50 | 7.27 | 15.20 | 8.88 |
| **Ours** | **79.48** | **75.46** | **80.36** | **77.09** | **63.38** | **48.11** | **70.23** | **61.40** |

out test volumes, achieving a mean AUC of 67.95. Using prompts derived from the same training set, we generate 500 synthetic CT volumes and fine-tune the classifier on the combined set.

Table 5 shows that training with synthetic data improves the mean AUC to 70.71, corresponding to a +2.76 absolute gain over the real-only baseline. Among the compared generators, the proposed approach achieves the best overall mean AUC, while performance remains label-dependent. Notable gains are observed on fine-grained findings such as Bronchiectasis, as well as on categories including Cardiomegaly and Emphysema where precise spatial cues are beneficial. These improvements align with the use of explicit mask guidance during synthesis, which encourages more localized and pathology-relevant signals for such conditions. The results also indicate complementary strengths across generators. GenerateCT performs better on Atelecta-

*Table 5.* Downstream multi-label classification AUC (%) on CT-RATE using synthetic training data from different generators. A classifier is trained on 500 real CT volumes and fine-tuned with an additional 500 generated volumes. The proposed method achieves the highest mean AUC, benefiting from disease-mask guidance that yields more localized and pathology-consistent signals. Best results are in **bold**.

| Test Set | Train Set | ArtCal | Atel | Cardio | CorCal | Emph | Hernia | Lymph | MedMat | Nodule | PeriEf |
|---|---|---|---|---|---|---|---|---|---|---|---|
| | Real | 71.53 | 55.67 | 76.73 | 69.72 | 66.23 | 58.27 | 71.00 | 75.44 | 61.39 | **79.42** |
| | +1xMedSyn | 74.65 | 67.66 | 76.58 | 70.18 | 62.55 | 63.99 | 74.50 | **81.27** | 63.98 | 77.32 |
| | +1xGenerateCT | 70.77 | **68.31** | 73.13 | 71.24 | 67.27 | 63.92 | **74.63** | 80.24 | **67.80** | 75.24 |
| | +1xOurs | **76.99** | 61.94 | **82.06** | **72.84** | **68.64** | **64.38** | 73.81 | 79.38 | 65.70 | 76.41 |
| **CT-RATE** | **Train Set** | **Bronch** | **Cons** | **FibSeq** | **Mosaic** | **Opacity** | **PeriTh** | **PleEf** | **SeptTh** | **Average** | |
| | Real | 52.32 | 69.06 | 65.74 | 63.38 | 66.69 | **64.14** | 79.13 | 77.25 | 67.95 | |
| | +1xMedSyn | 50.03 | 71.06 | 59.69 | **63.82** | 75.55 | 60.61 | **83.41** | 77.77 | 69.70 (+1.75) | |
| | +1xGenerateCT | 53.63 | **72.59** | 61.55 | 55.46 | 74.84 | 62.64 | 82.70 | **80.47** | 69.80 (+1.85) | |
| | +1xOurs | **56.99** | 70.00 | **66.09** | 61.72 | 72.09 | 63.78 | 82.61 | 77.25 | **70.71 (+2.76)** | |

*Abbreviations:* ArtCal = Arterial wall calcification; Atel = Atelectasis; Bronch = Bronchiectasis; Cardio = Cardiomegaly; Cons = Consolidation; CorCal = Coronary artery wall calcification; Emph = Emphysema; FibSeq = Pulmonary fibrotic sequela; Hernia = Hiatal hernia; Lymph = Lymphadenopathy; MedMat = Medical material; Mosaic = Mosaic attenuation pattern; Nodule = Lung nodule; Opacity = Lung opacity; PeriEf = Pericardial effusion; PeriTh = Peribronchial thickening; PleEf = Pleural effusion; SeptTh = Interlobular septal thickening.

*Table 6.* Ablation of the Foundation VAE on CT generation. Only the VAE is swapped; the diffusion backbone, training schedule, and conditioning are fixed.

| Split | VAE | FID↓ | CT-CLIP↑ |
|---|---|---|---|
| Normal | MedVAE | 11.28 | 20.76 |
| | Foundation VAE | **2.19** | **59.35** |
| Disease | MedVAE | 10.54 | 15.32 |
| | Foundation VAE | **4.78** | **51.49** |

sis and Nodule, while MedSyn achieves the highest AUC on Opacity and Medical material. This variability suggests that augmentation effectiveness depends on the target finding and motivates future work on condition-specific or targeted synthesis to maximize downstream benefits.

### 4.6. Ablation Study

**Contribution of the Foundation VAE.** The CT generation pipeline comprises a frozen VAE encoder–decoder and a separately trained diffusion model. To verify that the observed improvements originate from the Foundation VAE, we conduct a controlled ablation: the Foundation VAE is replaced with MedVAE (Varma et al., 2025a) while the diffusion backbone, training schedule, and all conditioning signals are kept identical.

As shown in Table 6, the Foundation VAE consistently outperforms MedVAE on both FID and CT-CLIP across normal and disease splits. FID drops from 11.28 to 2.19 on the normal split and from 10.54 to 4.78 on the disease split, while CT-CLIP improves by over 30 points in both cases. The gap is particularly pronounced in semantic alignment, where CT-CLIP nearly triples when MedVAE is replaced with the Foundation VAE, indicating that a higher-quality latent space preserves more clinically relevant features for the diffusion model to exploit.

**Effect of Conditioning Signals.** Conditioning signals used during training and inference are ablated under three settings. The first setting conditions only on organ masks, isolating explicit pathology mask supervision. The second setting introduces minimal pathology conditioning by using a single disease mask paired with a single-finding prompt. The finding text is taken from ReXGroundingCT. For cases containing multiple disease masks, one CT volume is generated per disease by pairing each mask with its corresponding single-finding prompt, and the metrics are averaged across diseases. The third setting uses multi-disease masks together with full report prompts, enabling compositional control over multiple abnormalities with richer textual descriptions.

The results are summarized in Table 7. Using organ masks alone provides anatomical grounding but yields weaker semantic alignment, with CT-CLIP of 43.40. Adding the single disease mask and single-finding prompt improves both coherence and semantic consistency, reducing FVD and increasing CT-CLIP, indicating that even sparse pathology conditioning helps localize abnormalities and stabilize generation. Using multi-disease masks together with full report prompts yields the strongest semantic alignment, achieving the highest CT-CLIP of 51.49. This setting increases FID compared with simpler conditioning, since multi-lesion synthesis and full report descriptions introduce greater appearance diversity.

## 5. Related work

### 5.1. Medical VAE

Robust vision encoders are foundational to medical imaging, where models must capture fine-grained anatomy and pathology while transferring across datasets and tasks. BTB3D (Hamamci et al., 2025) proposes a causal-convolutional encoder–decoder for text-to-CT synthesis and

*Table 7.* Ablation study on various conditioning signals. Best results are in **bold**.

| Organ mask | Disease mask | Text prompt | $FVD_{CT-CLIP}$ ↓ | $FID_{Avg}$ ↓ | $CT-CLIP_{Avg}$ ↑ |
|:---:|:---:|:---:|:---:|:---:|:---:|
| ✓ | – | Multi | 0.5172 | **3.72** | 43.40 |
| ✓ | Single | Single | **0.4805** | 3.76 | 46.14 |
| ✓ | Multi | Multi | 0.5088 | 4.35 | **51.49** |

long-context CT VLM, producing anatomically consistent CT volumes. In parallel, large medical autoencoders such as MedVAE (Varma et al., 2025b) demonstrate that scalable VAEs trained on medical corpora yield generalizable features for interpretation and downstream analysis. Within CT generation pipelines, several works *retrain* a VAE on CT as the vision feature extractor (e.g., MAISI (Guo et al., 2025b), GenerateCT (Hamamci et al., 2023)), which is effective but introduces a dedicated pretraining stage and increases data and compute costs. In this paper, we systematically study a *free-lunch* alternative: reusing *off-the-shelf* natural-video VAEs as CT encoders/decoders *without* any medical fine-tuning. Evaluating seven recent video VAEs, we find that their latent spaces already capture sufficient anatomical structure to support high-fidelity CT reconstruction and strong downstream segmentation. This *eliminates* the CT-specific VAE pretraining stage and markedly reduces the cost and complexity of CT generation.

## 5.2. Controllable CT Generation

Controllable CT generation (Chen et al., 2024b) aims to synthesize anatomically coherent 3D scans while allowing precise control over semantic and spatial factors. Early GAN-based methods (Mendes et al., 2023; Pesaranghader et al., 2021; Wu et al., 2024) offered limited controllability and mainly handled coarse modality translation. Text-conditioned diffusion frameworks (Xu et al., 2023; Hamamci et al., 2023; Guo et al., 2025a; Li et al., 2024) introduced semantic control through radiology reports, but they lack explicit spatial grounding and often place abnormalities at anatomically implausible locations. Recent works incorporate anatomical or tumor masks (Guo et al., 2024; Hu et al., 2023; Chen et al., 2024a) to improve geometric alignment, yet these approaches remain restricted to single-pathology synthesis and do not generalize across diverse disease types. Despite advances in realism, semantic control, and anatomy-aware modeling, existing methods still fail to unite text-level guidance with spatially grounded multi-disease synthesis.

## 6. Discussion

This work investigates controllable 3D chest CT synthesis conditioned on radiology text and mask-based spatial priors. Quantitative and qualitative results show that the proposed conditioning enables spatial relocation of lesions to specified regions and morphology changes across disease categories, while preserving organ geometry and inter-slice continuity.

Several limitations remain. First, controllability depends on conditioning mask quality: noisy boundaries, missing regions, or coarse pseudo masks can cause boundary artifacts and over-smoothed lesions, particularly for small findings (e.g., micronodules) and subtle textures (e.g., ground-glass), as illustrated in Appendix Figure 9. Second, rare disease categories and long-tail co-occurrences remain challenging, where synthesis quality degrades for atypical locations or severe distortions. On the evaluation side, automated metrics are sensitive to preprocessing and segmentation model bias, while expert assessment is limited in scale. Training data may also encode institution-specific characteristics, meaning that domain shift across scanners can affect both fidelity and downstream gains. Reported improvements should therefore be interpreted as evidence of feasibility rather than a guarantee of universal performance.

These limitations can be addressed by improving mask–text alignment, incorporating uncertainty-aware conditioning, and modeling multi-resolution anatomy-to-lesion interactions. Better handling of long-tail diseases may benefit from class-balanced objectives or retrieval-augmented conditioning. Extending the framework to longitudinal synthesis and multimodal clinical context (e.g., time-stamped reports, EMR features) could further support progression modeling. Finally, responsible deployment requires safeguards against misuse and clear disclosure that synthesized images are intended for research augmentation rather than direct clinical decision-making.

## 7. Conclusion

We show that Foundation VAEs pretrained on natural images and videos can be reused as a practical CT representation interface without medical fine-tuning. Across seven existing video VAEs, frozen encoder and decoder reconstructions preserve anatomy and maintain downstream segmentation accuracy, while reconstruction-based augmentation improves robustness on boundary-sensitive metrics. Using the same fixed latent space, a conditional latent diffusion model enables controllable 3D CT generation with joint conditioning on organ masks, disease masks, and radiology reports, supporting multiple disease types within a single generator. Collectively, these findings establish Foundation VAEs as a scalable, data-efficient, and generalizable representation paradigm for CT, streamlining system design while reducing the need for domain-specific architectural choices across heterogeneous clinical distributions.

## Impact Statement

This work enables training-free reuse of large pretrained VAEs for 3D CT reconstruction, augmentation, and con-

trollable generation, with the goal of lowering compute and engineering cost and improving robustness for downstream medical imaging research. Reconstruction-based augmentation can help reduce dependence on additional annotations, and mask- and text-conditioned generation can support benchmarking and controlled studies of model behavior. Risks include misuse of synthetic CT volumes, amplification of dataset biases, and misleading conclusions if synthetic data are treated as fully representative of clinical distributions. Generated samples may contain hallucinated or missing findings that can distort evaluation. Mitigation includes clear labeling of all synthetic outputs, restricting use to research and benchmarking, validating conclusions on real clinical data, and reporting failure cases and limitations. The approach is not intended for diagnostic or clinical decision-making.

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

# A. Theoretical Analysis

Training high quality 3D CT generative models is computationally demanding. A single volumetric scan often contains hundreds of high resolution slices (e.g., $512\times512\times200+$), and end to end optimization must maintain coherent 3D structure across the stack. Latent diffusion alleviates this cost by learning in a compressed representation; nevertheless, many CT pipelines still pretrain a CT specific autoencoder before training the diffusion model. This introduces additional engineering effort (architecture design, compression tuning, training stability) and extra compute, and it couples the encoder decoder pair to a particular medical dataset.

A key observation is that reconstruction training is largely semantics agnostic. The autoencoder is optimized to invert inputs and preserve local geometry and texture, rather than to recognize whether the volume comes from natural videos or CT scans. As a result, a strong autoencoder trained on large scale natural data may still retain task relevant structure when applied to CT volumes, even without medical domain fine tuning. This motivates the following question: *when can an off the shelf natural video VAE be reused as a CT encoder decoder while keeping downstream performance nearly unchanged?*

**Setup.** Let $(x, y) \sim P$, where $x \in \mathcal{X}$ denotes a CT volume and $y \in \mathcal{Y}$ denotes its voxel-wise annotation. A pretrained VAE induces a reconstruction operator

$$T(x) := D(E(x)), \tag{9}$$

with encoder $E$ and decoder $D$. Let $f_\theta$ be a downstream predictor trained with loss $\ell(\cdot, \cdot)$. We define the population risks

$$\mathcal{R}_P(\theta) := \mathbb{E}_{(x,y)\sim P} \left[ \ell\left( f_\theta(x), y \right) \right], \tag{10}$$

and

$$\mathcal{R}_{T_\sharp P}(\theta) := \mathbb{E}_{(x,y)\sim P} \left[ \ell\left( f_\theta(T(x)), y \right) \right], \tag{11}$$

where $T_\sharp P$ denotes the pushforward distribution induced by reconstruction.

**Assumption: task relevant reconstruction stability.** There exists a feature mapping $\phi : \mathcal{X} \to \mathbb{R}^d$ that captures task relevant geometric structure such that

$$\mathbb{E}_{x\sim P_X}[\|\phi(x) - \phi(T(x))\|_2] \leq \varepsilon_\phi. \tag{12}$$

Moreover, the loss is Lipschitz with respect to $\phi$, meaning for all $x, x', y, \theta$,

$$\left|\ell(f_\theta(x), y) - \ell(f_\theta(x'), y)\right| \leq L_\ell \left\|\phi(x) - \phi(x')\right\|_2. \tag{13}$$

This assumption requires preservation of task relevant structure, rather than perfect pixel fidelity.

**Theorem 1: risk gap induced by reconstruction.** Let

$$\hat{\theta} \in \arg\min_\theta \mathcal{R}_{T_\sharp P}(\theta), \qquad \theta^\star \in \arg\min_\theta \mathcal{R}_P(\theta). \tag{14}$$

Then the excess risk satisfies

$$\mathcal{R}_P(\hat{\theta}) - \mathcal{R}_P(\theta^\star) \leq 2L_\ell\varepsilon_\phi. \tag{15}$$

**Proof.** For any $\theta$ and any $(x, y)$, applying (18) with $x' = T(x)$ gives

$$\ell(f_\theta(x), y) \leq \ell(f_\theta(T(x)), y) + L_\ell|\phi(x) - \phi(T(x))|_2. \tag{16}$$

Swapping the roles of $x$ and $T(x)$ yields

$$\ell(f\theta(T(x)), y) \leq \ell(f_\theta(x), y) + L_\ell|\phi(x) - \phi(T(x))|_2. \tag{17}$$

Taking expectation over $(x, y) \sim P$ leads to the uniform bound

$$\left|\mathcal{R}_P(\theta) - \mathcal{R}_{T\sharp P}(\theta)\right| \leq L_\ell\varepsilon_\phi, \tag{18}$$

where we used (18). Now apply (18) to $\theta = \hat{\theta}$:

$$\mathcal{R}_P(\hat{\theta}) \leq \mathcal{R}_{T_\sharp P}(\hat{\theta}) + L_\ell\varepsilon_\phi. \tag{19}$$

By optimality of $\hat{\theta}$ under $T_\sharp P$,

$$\mathcal{R}_{T_\sharp P}(\hat{\theta}) \leq \mathcal{R}_{T_\sharp P}(\theta^\star). \tag{20}$$

Apply (18) again to $\theta = \theta^\star$:

$$\mathcal{R}_{T_\sharp P}(\theta^\star) \leq \mathcal{R}_P(\theta^\star) + L\ell\varepsilon_\phi. \tag{21}$$

Combining the three inequalities gives

$$\mathcal{R}_P(\hat{\theta}) \leq \mathcal{R}_P(\theta^\star) + 2L_\ell\varepsilon_\phi, \tag{22}$$

which proves (18). □

### A.1. Empirical verification

Equation (22) indicates that reusing an off-the-shelf VAE is justified when the task-relevant distortion $\varepsilon_\phi$ is small. Since $\varepsilon_\phi$ is defined in a latent feature space and is not directly interpretable, we adopt a more intuitive proxy based on *segmentation consistency*. Specifically, we measure whether a fixed CT segmenter produces stable predictions on the original volume $x$ and its reconstruction $T(x)$.

Let $g(\cdot)$ be a frozen pretrained 3D CT segmentation network, and define

$$\hat{y}_i := g(x_i), \qquad \tilde{y}_i := g(T(x_i)), \tag{23}$$

where $\hat{y}_i$ and $\tilde{y}_i$ denote the predicted masks (after thresholding or argmax) on the original and reconstructed volumes, respectively. We then compute Dice and Normalized Surface Dice (NSD) between the two predictions:

$$\text{Dice}(a, b) := \frac{2|a \cap b|}{|a| + |b|}, \tag{24}$$

$$\text{NSD}_\tau(a, b) := \frac{1}{|\partial a|} \sum_{u \in \partial a} \mathbf{1}(\text{dist}(u, \partial b) \leq \tau), \tag{25}$$

where $\partial a$ denotes the surface of mask $a$, $\text{dist}(\cdot, \partial b)$ is the distance to the surface of $b$, and $\tau$ is the tolerance (in voxels or millimeters).

Finally, we define an interpretable empirical proxy for reconstruction distortion as

$$\hat{\varepsilon}_\phi := 1 - \frac{1}{n} \sum_{i=1}^{n} \frac{\text{Dice}(\hat{y}_i, \tilde{y}_i) + \text{NSD}_\tau(\hat{y}_i, \tilde{y}_i)}{2}. \tag{26}$$

A smaller $\hat{\varepsilon}_\phi$ means higher segmentation consistency, indicating that reconstruction preserves task-relevant geometry.

Across evaluated video VAEs, we observe consistently high segmentation consistency between $x$ and $T(x)$, and downstream models trained on reconstructed CTs achieve accuracy comparable to training on original CTs. Together, these results support the conclusion that off-the-shelf natural-video VAEs can serve as effective CT encoders and decoders without medical-domain fine-tuning.

## B. VAEs evaluated in this work

We consider seven publicly released video VAEs as candidate *Foundation VAE* backbones and apply them to 3D CT volumes without medical adaptation.

- **WAN2.1** (Wan et al., 2025) and **WAN2.2** (Wan et al., 2025): video autoencoders released with the Wan video model family, used as latent encoders and decoders for large scale video generation.

- **VideoVAE+** (Xing et al., 2024): a cross modal video VAE designed for large motion video autoencoding.

- **IVVAE** (Wu et al., 2025): an improved video VAE that strengthens spatiotemporal reconstruction fidelity for latent generative models.

- **CVVAE** (Zhao et al., 2024): a compatible video VAE that targets stable and diffusion friendly latent spaces for generative video models.

- **WFVAE** (Li et al., 2025): a wavelet guided video VAE that improves high frequency detail preservation via energy flow modeling.

- **LeanVAE** (Cheng & Yuan, 2025): an ultra efficient wavelet based video VAE that trades minimal compute for strong reconstruction quality.

We additionally report results for models trained on medical data, serving as in domain references.

- **MedVAE** (Varma et al., 2025b): a large scale medical autoencoder trained on diverse medical imagery for transferable representations.

- **MAISI** (Guo et al., 2025b): a CT synthesis framework that includes a CT encoder and decoder paired with a latent diffusion generator.

*Table 8.* Comparison of latent representation sizes across VAEs. "Relative Latent Size" is normalized to MedVAE. Although models differ in compression ratio and latent channel count, their total latent sizes lie within the same order of magnitude. Notably, VideoVAE+, CVVAE, and LeanVAE match MedVAE in total latent size, yet consistently outperform it on segmentation and generation tasks, suggesting that the gains stem from the quality of the latent interface rather than from a larger latent representation.

| Model | Comp. Ratio $(X \times Y \times Z)$ | Latent Channels $(C)$ | Latent Dimensions $(C \times H/f \times W/f \times D/f)$ | Total Latent Elements | Relative Size vs. MedVAE |
|---|---|---|---|---|---|
| WAN2.1 | $8 \times 8 \times 4$ | 16 | $16 \times 64 \times 64 \times 4$ | 262,144 | $4.0\times$ |
| WAN2.2 | $16 \times 16 \times 4$ | 48 | $48 \times 32 \times 32 \times 4$ | 196,608 | $3.0\times$ |
| VideoVAE+ | $8 \times 8 \times 4$ | 4 | $4 \times 64 \times 64 \times 4$ | 65,536 | $1.0\times$ |
| IVVAE | $8 \times 8 \times 4$ | 16 | $16 \times 64 \times 64 \times 4$ | 262,144 | $4.0\times$ |
| CVVAE | $8 \times 8 \times 4$ | 4 | $4 \times 64 \times 64 \times 4$ | 65,536 | $1.0\times$ |
| WFVAE | $8 \times 8 \times 4$ | 8 | $8 \times 64 \times 64 \times 4$ | 131,072 | $2.0\times$ |
| LeanVAE | $8 \times 8 \times 4$ | 4 | $4 \times 64 \times 64 \times 4$ | 65,536 | $1.0\times$ |
| MedVAE | $4 \times 4 \times 4$ | 1 | $1 \times 128 \times 128 \times 4$ | 65,536 | $1.0\times$ |
| MAISI | $4 \times 4 \times 4$ | 4 | $4 \times 128 \times 128 \times 4$ | 262,144 | $4.0\times$ |

## C. Datasets and Implementation Details

CT-RATE (Hamamci et al., 2024) is our primary dataset. The diseased subset is defined by its overlap with ReXGroundingCT (Baharoon et al., 2025). The train/test split is re-defined because the original ReXGroundingCT split leaves some disease categories unrepresented in the test set; a strict patient-level split is enforced to prevent any leakage. In total, the generation task comprises 4,961 training cases and 80 validation cases across 18 disease categories (Table 9).

Organ masks are obtained by combining VISTA3D (He et al., 2025) (127-class organ and lesion segmentation) with TotalSegmentator (Wasserthal et al., 2023) (body- and vessel-level masks) to produce whole-body anatomical segmentation. Tumor-related labels from VISTA3D are removed, and body and vessel masks are added, resulting in 126 classes in total. Label indices follow the original VISTA3D label definition. The full list of labels is provided in Table 10.

For the downstream classification task, we further sample 500 training volumes from the generative model training set. The synthetic training data use the same text prompts, organ masks, and disease masks. We also sample 200 validation volumes from the original CT-RATE dataset, as disease masks are not required for classification.

All models are implemented in PyTorch and MONAI (Cardoso et al., 2022). Generative inference is performed on NVIDIA B200 GPUs, and downstream evaluations are conducted on NVIDIA A100 GPUs. Evaluation scripts follow the VLM3D Challenge toolkit `https://github.com/forithmus/VLM3D-Dockers`.

*Table 9.* Disease distribution for generative model and downstream task splits.

| Finding | Generative model | | Downstream task | |
|---|---|---|---|---|
| | Train | Test | Train | Test |
| Normal | 2,395 | 30 | 103 | 23 |
| Arterial wall calcification | 308 | 4 | 51 | 87 |
| Atelectasis* | 571 | 11 | 90 | 75 |
| Bronchiectasis* | 234 | 3 | 36 | 27 |
| Cardiomegaly | 111 | 3 | 20 | 40 |
| Consolidation* | 490 | 7 | 71 | 64 |
| Coronary artery wall calcification | 291 | 7 | 55 | 81 |
| Emphysema* | 411 | 9 | 71 | 58 |
| Hiatal hernia | 9 | 0 | 1 | 37 |
| Interlobular septal thickening* | 154 | 3 | 29 | 24 |
| Lung nodule* | 1301 | 31 | 207 | 119 |
| Lung opacity* | 1096 | 21 | 161 | 120 |
| Lymphadenopathy | 475 | 9 | 73 | 82 |
| Medical material | 195 | 4 | 40 | 27 |
| Mosaic attenuation pattern | 101 | 1 | 11 | 22 |
| Peribronchial thickening | 184 | 3 | 29 | 33 |
| Pericardial effusion | 116 | 4 | 29 | 26 |
| Pleural effusion* | 200 | 4 | 45 | 43 |
| Pulmonary fibrotic sequela | 574 | 10 | 87 | 73 |
| Total | 4,961 | 80 | 500 | 200 |

*Note:* Findings marked with * have disease masks from the ReXGroundingCT (Baharoon et al., 2025).

*Table 10.* Organ label indices used in this work (124 classes, excluding background).

| Structure | Label | Structure | Label | Structure | Label |
|---|---|---|---|---|---|
| liver | 1 | spleen | 3 | pancreas | 4 |
| right kidney | 5 | aorta | 6 | inferior vena cava | 7 |
| right adrenal gland | 8 | left adrenal gland | 9 | gallbladder | 10 |
| esophagus | 11 | stomach | 12 | duodenum | 13 |
| left kidney | 14 | bladder | 15 | portal vein and splenic vein | 17 |
| small bowel | 19 | brain | 22 | pancreatic tumor | 24 |
| hepatic vessel | 25 | hepatic tumor | 26 | colon cancer primaries | 27 |
| left lung upper lobe | 28 | left lung lower lobe | 29 | right lung upper lobe | 30 |
| right lung middle lobe | 31 | right lung lower lobe | 32 | vertebrae L5 | 33 |
| vertebrae L4 | 34 | vertebrae L3 | 35 | vertebrae L2 | 36 |
| vertebrae L1 | 37 | vertebrae T12 | 38 | vertebrae T11 | 39 |
| vertebrae T10 | 40 | vertebrae T9 | 41 | vertebrae T8 | 42 |
| vertebrae T7 | 43 | vertebrae T6 | 44 | vertebrae T5 | 45 |
| vertebrae T4 | 46 | vertebrae T3 | 47 | vertebrae T2 | 48 |
| vertebrae T1 | 49 | vertebrae C7 | 50 | vertebrae C6 | 51 |
| vertebrae C5 | 52 | vertebrae C4 | 53 | vertebrae C3 | 54 |
| vertebrae C2 | 55 | vertebrae C1 | 56 | trachea | 57 |
| left iliac artery | 58 | right iliac artery | 59 | left iliac vena | 60 |
| right iliac vena | 61 | colon | 62 | left rib 1 | 63 |
| left rib 2 | 64 | left rib 3 | 65 | left rib 4 | 66 |
| left rib 5 | 67 | left rib 6 | 68 | left rib 7 | 69 |
| left rib 8 | 70 | left rib 9 | 71 | left rib 10 | 72 |
| left rib 11 | 73 | left rib 12 | 74 | right rib 1 | 75 |
| right rib 2 | 76 | right rib 3 | 77 | right rib 4 | 78 |
| right rib 5 | 79 | right rib 6 | 80 | right rib 7 | 81 |
| right rib 8 | 82 | right rib 9 | 83 | right rib 10 | 84 |
| right rib 11 | 85 | right rib 12 | 86 | left humerus | 87 |
| right humerus | 88 | left scapula | 89 | right scapula | 90 |
| left clavicula | 91 | right clavicula | 92 | left femur | 93 |
| right femur | 94 | left hip | 95 | right hip | 96 |
| sacrum | 97 | left gluteus maximus | 98 | right gluteus maximus | 99 |
| left gluteus medius | 100 | right gluteus medius | 101 | left gluteus minimus | 102 |
| right gluteus minimus | 103 | left autochthon | 104 | right autochthon | 105 |
| left iliopsoas | 106 | right iliopsoas | 107 | left atrial appendage | 108 |
| brachiocephalic trunk | 109 | left brachiocephalic vein | 110 | right brachiocephalic vein | 111 |
| left common carotid artery | 112 | right common carotid artery | 113 | costal cartilages | 114 |
| heart | 115 | left kidney cyst | 116 | right kidney cyst | 117 |
| prostate | 118 | pulmonary vein | 119 | skull | 120 |
| spinal cord | 121 | sternum | 122 | left subclavian artery | 123 |
| right subclavian artery | 124 | superior vena cava | 125 | thyroid gland | 126 |
| vertebrae S1 | 127 | airway | 132 | vessel | 133 |
| body | 200 | | | | |

## D. Multi-disease Synthesis Results

We provide additional qualitative results for synthesized chest CT volumes (Figure 7 and Figure 8). The left column shows real CT, and the right column shows our generated CT. For each case, we visualize three consecutive slices at the same z locations in both volumes, where different colors indicate different disease findings.

For the same disease category, the model accurately generates lesions at different specified locations according to the input text and masks. Across different diseases, the model produces distinct and pathology-consistent morphologies and texture details, indicating strong disease-specific controllability. Meanwhile, anatomical structures remain well preserved, with realistic organ appearance and spatial coherence. Three-view comparisons in Figure 10 further indicate superior z-axis continuity and the most faithful anatomical preservation.

Failure cases are shown in Figure 9. In multi-mask settings, the model may ignore one of the input masks, leading to incomplete spatial control. In addition, synthesizing small-scale findings, such as pulmonary nodules, remains challenging.

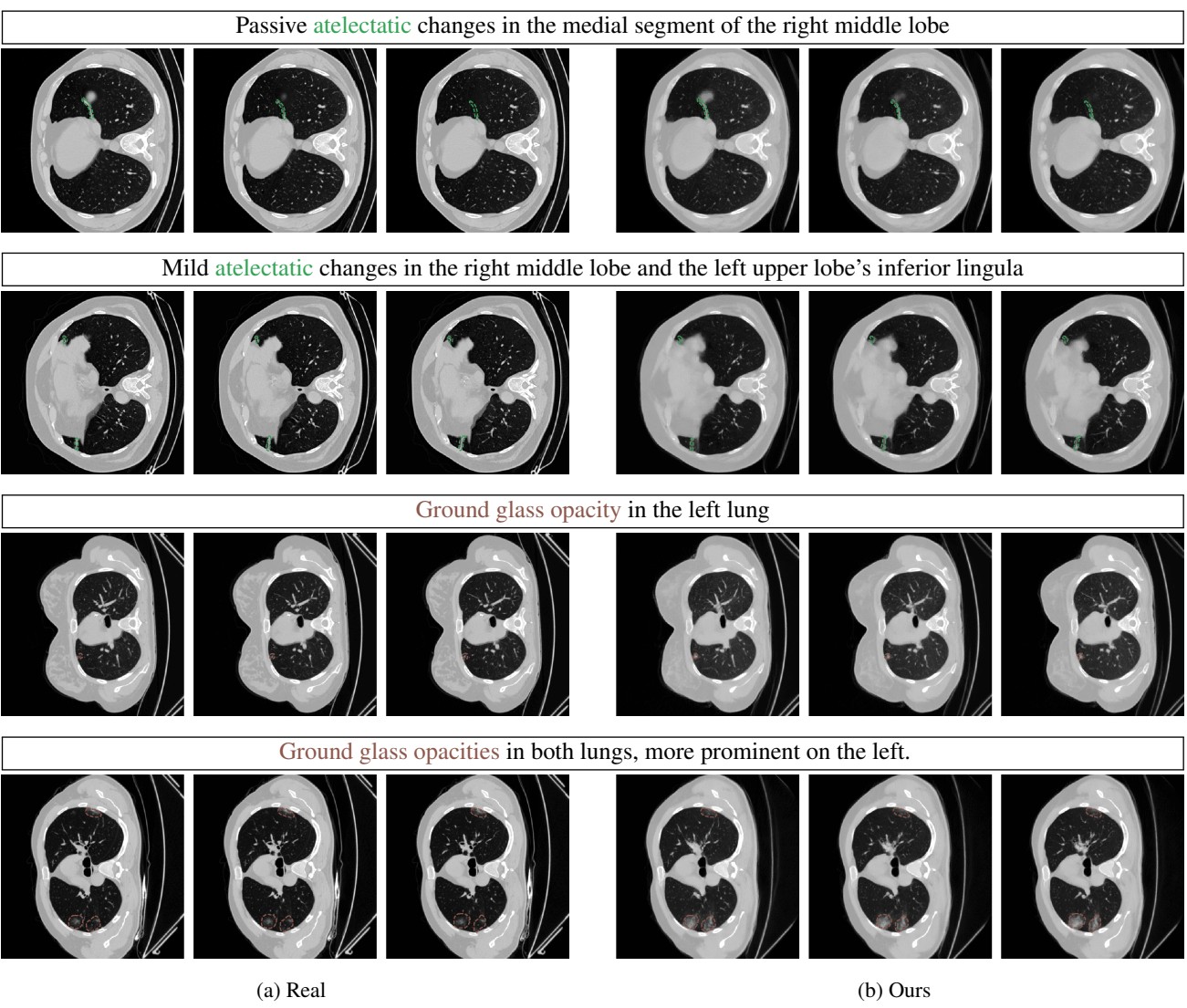

(a) Real                                (b) Ours

*Figure 7.* Qualitative comparison between real (left) and ours (right) across representative slices. Color text highlights key findings in the report.

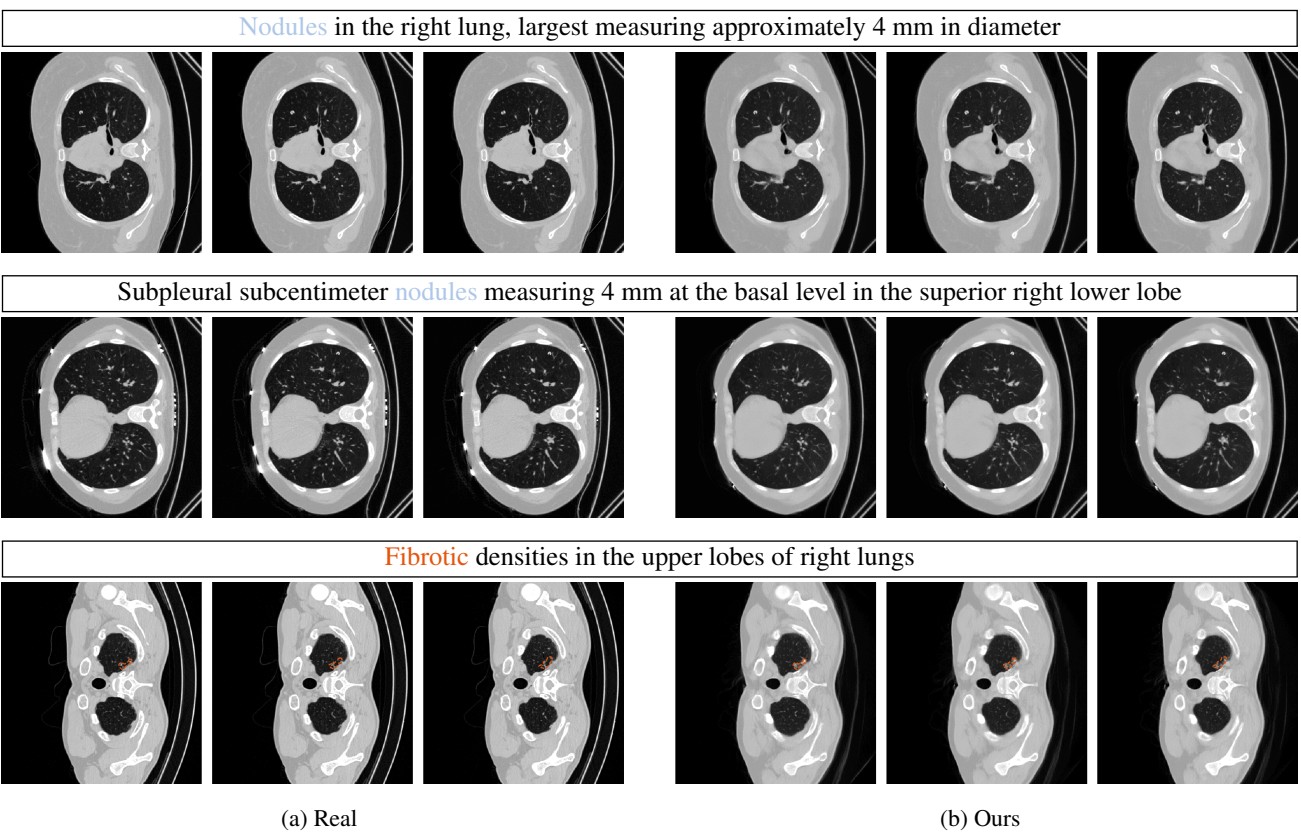

*Figure 8.* Qualitative comparison between real (left) and ours (right) across representative slices. Color text highlights key findings in the report.(Cont.)

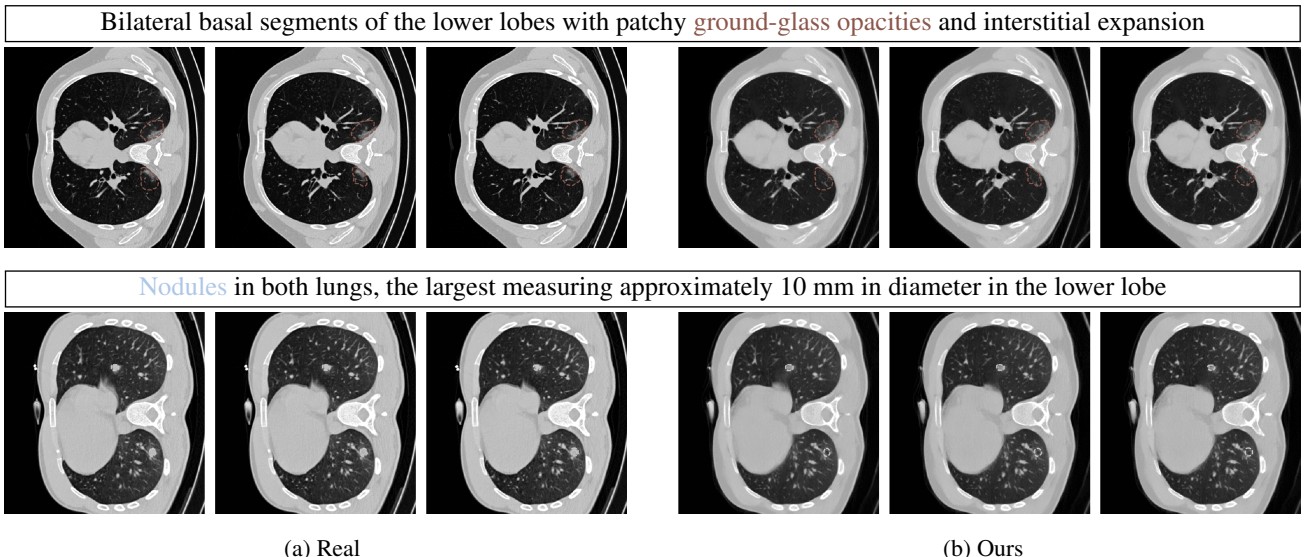

*Figure 9.* Failure cases: qualitative comparisons of representative slices between real CT (left) and ours (right). Colored text highlights key findings in the report.

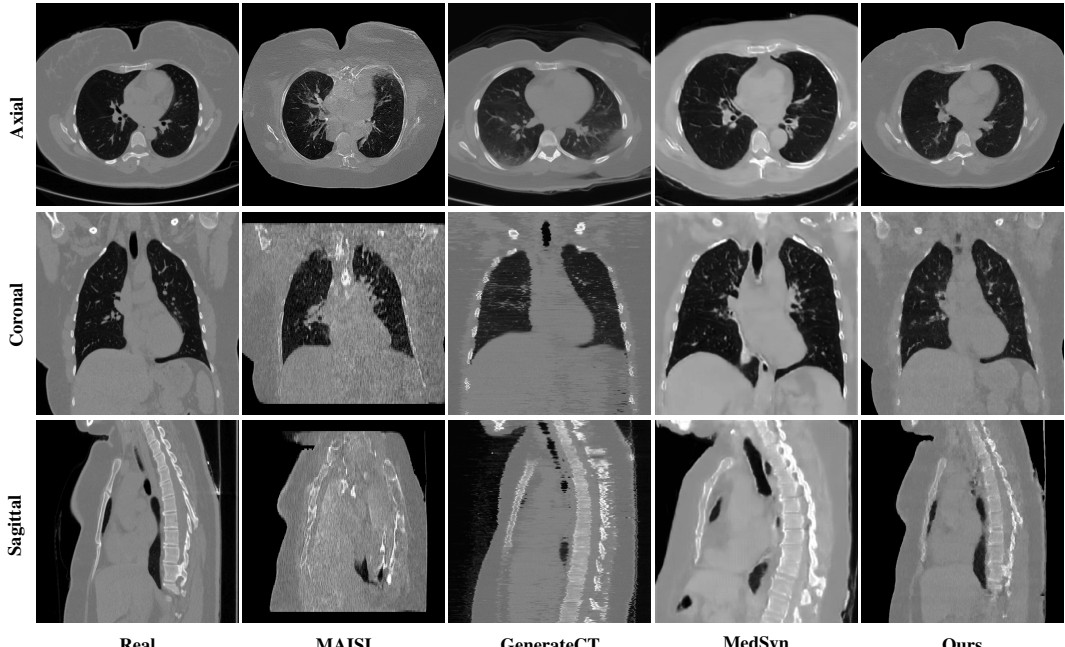

*Figure 10.* Three-view qualitative comparison between real CT and generative models.

