# OpenReview forum: "Foundation VAE for CT Reconstruction, Augmentation, and Generation"
_ICML.cc/2026/Conference — ICML 2026 regular_

### Official Review · Reviewer_Cf9T · 2026-03-10

**Soundness:** 3
**Presentation:** 3
**Significance:** 3
**Originality:** 2
**Overall Recommendation:** 4
**Confidence:** 3

**Summary:**

This paper studies whether large-scale pretrained video VAEs can be reused for 3D CT reconstruction, augmentation, and generation without medical-domain fine-tuning. The work explores the idea of treating a pretrained “foundation VAE” as a frozen representation interface and training diffusion models in its latent space for CT synthesis.

**Compliance With Llm Reviewing Policy:**

Affirmed.

**Key Questions For Authors:**

1 Strengths. The idea of “training-free” representation reuse is practically valuable and well motivated. The experimental evaluation is relatively comprehensive, covering reconstruction, segmentation, anatomical alignment, and downstream classification tasks. The results show that frozen foundation VAEs can effectively preserve anatomical structures and support multi-disease generation within a unified latent space.
2 Limited novelty. The generation framework largely follows a standard latent diffusion architecture, using mask concatenation and cross-attention for text conditioning. The proposed 3D consistency module is incremental in nature. Therefore, the main contribution lies more in system integration and paradigm validation rather than in algorithmic innovation.
3 Issues in presentation and analysis. The term “training-free” is somewhat overstated, since a diffusion model is still trained on CT data. The theoretical analysis relies on stability assumptions that are not sufficiently validated empirically. Additional discussion of failure cases (e.g., subtle lesions, cross-domain generalization, pathological hallucinations) would further strengthen the rigor of the paper.
Overall, the paper presents solid experimental results and clear practical significance, but the level of methodological innovation is moderate.

**Limitations:**

Yes.

**Strengths And Weaknesses:**

The main strength of the paper is that it addresses a practically relevant problem. Training domain-specific generative models for medical imaging is expensive and data-intensive, so the idea of reusing pretrained foundation models is appealing. The framework is conceptually simple and unified, enabling CT reconstruction, reconstruction-based augmentation, and conditional CT generation within the same latent space. The experiments are reasonably comprehensive and include reconstruction quality, segmentation performance, and controllable generation results.
However, the methodological novelty is somewhat limited. The overall pipeline largely follows standard latent diffusion modeling, and the conditioning mechanism using mask concatenation and text cross-attention is relatively conventional. As a result, the main contribution is closer to empirical validation of a paradigm rather than introducing a new algorithmic technique. In addition, the “training-free” claim is somewhat overstated since diffusion models are still trained on CT data. The paper would also benefit from deeper analysis of failure cases and stronger discussion of limitations.
Overall, the work provides interesting empirical observations about representation reuse in medical generative modeling, but the novelty and technical depth are moderate.

---

> ### Author Rebuttal · Authors · 2026-03-31
>
> > **Q1.** *Limited novelty. The generation framework largely follows a standard latent diffusion architecture, using mask concatenation and cross-attention for text conditioning. The proposed 3D consistency module is incremental in nature. Therefore, the main contribution lies more in system integration and paradigm validation rather than in algorithmic innovation.*
>
> - We agree that the paper is not primarily about introducing a radically new latent diffusion architecture, and that part of its contribution lies in system integration and paradigm validation. However, we respectfully believe this does not imply limited novelty. In machine learning, some of the most influential works are valuable not because they introduce a large architectural change, but because they revisit a widely held assumption, validate a new design principle at scale, and provide actionable guidance for the community. Canonical examples include work on ImageNet pretraining and scaling laws, whose impact comes largely from clarifying when existing components can be reused or scaled effectively, rather than from proposing an entirely new backbone.
> - In this paper, the key scientific contribution is precisely of this form. We study a question that is both practically important and previously underexplored: is CT-specific VAE training actually necessary for useful 3D medical latent modeling? Our results suggest that a fully frozen foundation VAE pretrained on natural images/videos can already serve as a strong latent interface for CT reconstruction, augmentation, and generation, while reducing medical data curation, engineering effort, and compute cost. We believe this finding is valuable to the community because it shifts attention from repeatedly training domain-specific medical VAEs toward a more scalable paradigm of foundation latent reuse.
> - We will revise the paper to make this positioning clearer. In particular, we will present the contribution less as an architecture-centric advance and more as a methodological and scientific finding: a systematic validation that frozen foundation representations can transfer surprisingly well to 3D CT, with clear practical implications for future medical generative modeling.
>
> > **Q2.** *Issues in presentation and analysis. The term “training-free” is somewhat overstated, since a diffusion model is still trained on CT data. The theoretical analysis relies on stability assumptions that are not sufficiently validated empirically. Additional discussion of failure cases (e.g., subtle lesions, cross-domain generalization, pathological hallucinations) would further strengthen the rigor of the paper.*
>
> Thank you for these helpful comments. We agree that the presentation can be made more precise, and we will revise the paper accordingly.
> - First, we will clarify the use of “training-free.” Our intent was to describe the Foundation VAE itself, which is used without any medical fine-tuning, rather than the entire CT generation pipeline, where the diffusion model is still trained on CT data. To avoid misunderstanding, we will use more precise wording in the revision.
> - Second, we will better position the theoretical analysis. Our goal is not to present a first-principles proof, but an assumption-based argument: if the Foundation VAE preserves task-relevant structure, then downstream segmentation risk can also be approximately preserved. In the current submission, the evidence is mainly indirect—reconstruction errors are largely noise-like, while downstream segmentation remains strong, especially on boundary-sensitive metrics. In the revised version, we will make this logic clearer and add more direct empirical support.
> - Third, we agree that a fuller discussion of failure cases would strengthen the paper. We already include related analysis in Appendix Fig. 9 and Section F, covering subtle lesion generation (e.g., tiny nodules), limitations on rare/long-tail disease patterns and cross-domain generalization, and representative cases of pathological hallucination. We will highlight these limitations more clearly in the main text.

---

> > ### Author Rebuttal · Reviewer_Cf9T · 2026-04-06
> >
> > In Q2, the authors clearly clarified the explicit meaning of “train-free,” and additional discussion of failure cases has also been supplemented in the appendix. For Q1, although the authors explained their claimed innovations, I still consider the novelty to be limited.

---

> > > ### Author Response · Authors · 2026-04-08
> > >
> > > Thank you for recognizing our rebuttal. We would like to explain again why we believe this paper provides a valuable contribution.
> > >
> > > 1. We agree that our paper does not propose a very new model architecture. However, we believe our work brings several new points for CT synthesis.
> > >
> > > - Previous CT synthesis methods are often complex, use multiple stages, and rely on specially trained VAEs, such as GenerateCT and MAISI. In contrast, we show that strong CT synthesis does not always need a complex design or high compute cost, as long as the latent space is strong enough. With our foundation VAE, a standard latent diffusion model, mask concatenation, cross attention for text, and a 3D consistency layer, we achieve state of the art generation results. These results are shown in Figure 4, Table 2, Figure 5, Table 3, the 3D demos in the supplementary material, and the new results in our response for Reviewer VXKc.
> > >
> > > - Previous CT synthesis works also study disease or lesion generation, but they support only a small number of diseases. Based on the CT RATE and ReXGroundingCT datasets, our work is the first to support 18 diseases. In earlier work, the largest number was 6 in MAISI. We also show that our generated data helps downstream tasks, as shown in Table 4. We believe this is important because it gives useful guidance for medical use of synthetic data.
> > >
> > > - Previous CT synthesis works usually use only one type of condition. For example, MedSyn uses class condition, GenerateCT and Text2CT use text condition, and MAISI uses mask condition. In contrast, our paper combines both text and mask conditions to improve generation quality, as shown in Table 5.
> > >
> > > 2. In machine learning, some important works matter not because they introduce a new architecture, but because they revisit a common belief, test a new idea at scale, and give useful guidance to the community. Examples include work on ImageNet pretraining [1], scaling laws [2,3,4], and video generation [5]. Their impact mostly comes from showing how existing methods can be reused or scaled well, not from creating a fully new backbone.
> > >
> > >    We believe our paper makes this kind of contribution. More broadly, we think studying transfer learning for VAEs is an important question for medical imaging. Our results show that a fully frozen foundation VAE, pretrained on natural images and videos, can already provide a strong latent space for CT reconstruction, augmentation, and generation, while reducing medical data collection, engineering effort, and compute cost. We believe this finding is useful for the community because it suggests a more scalable direction: reusing foundation latents instead of repeatedly training domain specific medical VAEs.
> > >
> > > [1] He K, et al. Rethinking ImageNet Pre training. ICCV 2019. Cited over 1,600 times.
> > >
> > > [2] Aghajanyan A, et al. Scaling laws for generative mixed modal language models. ICML 2023.
> > >
> > > [3] Clark A, et al. Unified scaling laws for routed language models. ICML 2022.
> > >
> > > [4] Alabdulmohsin I M, et al. Revisiting neural scaling laws in language and vision. NeurIPS 2022.
> > >
> > > [5] Bai, Jianhong, et al. Recammaster: Camera-controlled generative rendering from a single video. ICCV 2025. Best Paper Finalist.

---

### Official Review · Reviewer_jB2K · 2026-03-11

**Soundness:** 3
**Presentation:** 3
**Significance:** 4
**Originality:** 3
**Overall Recommendation:** 4
**Confidence:** 4

**Summary:**

The paper “Foundation VAE for CT Reconstruction, Augmentation, and Generation” presents a simple yet interesting idea. Instead of repeatedly training large VAEs specialized for medical images from scratch in order to enable latent diffusion models, the authors propose to leverage large Foundation VAEs that have been pretrained on a broad corpus of images. Through several experiments, the paper shows that these Foundation VAEs already provide a sufficiently expressive latent space to allow effective CT image generation by simply training a latent diffusion model on top of the pretrained representation.

**Compliance With Llm Reviewing Policy:**

Affirmed.

**Key Questions For Authors:**

No questions

**Limitations:**

Already discussed

**Strengths And Weaknesses:**

Overall, the paper is well written and the idea is appealing. I only have a few relatively minor observations that the authors may consider addressing.

1. Some terms used in the paper appear in the literature with multiple possible meanings, and it would therefore be helpful to clarify their intended usage in this context. For example, the term “VAE Reconstruction” (line 107) may refer either to the output $D(E(x))$ of the autoencoder, or to the solution of an inverse problem where a VAE is used as a prior. Similarly, the term “CT Reconstruction” (line 139) may refer either to the solution of a CT inverse problem or simply to the decoded output $D(E(x))$ when the model is applied to CT data.

2. In the description of the model architecture, a few implementation details could be clarified further. For instance, it would be helpful to explain how cross-attention (line 212) is used to inject textual conditioning into the denoising U-Net architecture, and at which stage of the pipeline the 3D consistency layer is applied.

3. Finally, I believe the comparison against the baselines could be reconsidered. In the data augmentation experiment, the classifier is first pretrained on real CT volumes and then fine-tuned on synthetic volumes. However, in many practical data augmentation settings the model is trained from scratch on a mixture of real and synthetic data. Evaluating the approach in such a setting might provide a more representative assessment of the benefits of the proposed method. In addition, the Related Work section mentions MedVAE, a large autoencoder trained specifically on medical images. It would be interesting to include a comparison with this approach in the experiments.

---

> ### Author Rebuttal · Authors · 2026-03-31
>
> > **Q1.** *Some terms used in the paper appear in the literature with multiple possible meanings, and it would therefore be helpful to clarify their intended usage in this context. For example, the term “VAE Reconstruction” (line 107) may refer either to the output of the autoencoder, or to the solution of an inverse problem where a VAE is used as a prior. Similarly, the term “CT Reconstruction” (line 139) may refer either to the solution of a CT inverse problem or simply to the decoded output  when the model is applied to CT data.*
>
> Thank you for the valuable suggestion. We agree that these terms can be ambiguous, and we will make the terminology consistent in the revised paper. Specifically, we will use CT Reconstruction throughout to refer only to the autoencoder output, i.e., the CT volume obtained after passing the input through the encoder and decoder, and will avoid wording that could be confused with CT inverse-problem reconstruction.
>
> > **Q2.** *In the description of the model architecture, a few implementation details could be clarified further. For instance, it would be helpful to explain how cross-attention (line 212) is used to inject textual conditioning into the denoising U-Net architecture, and at which stage of the pipeline the 3D consistency layer is applied.*
>
> - We will clarify these implementation details in the revised version.
> In the cross-attention module, the flattened image features are used as the queries, while the text report embeddings are used as the keys and values, allowing the denoising U-Net to condition image generation on the report content.
> - The 3D consistency layer is applied at every block of the denoising U-Net, as illustrated in Fig. 3. Each block consists of a ResBlock, a spatial attention layer, and a 3D consistency layer, so cross-slice consistency is enforced throughout the denoising process rather than only at a single stage.
>
> > **Q3.** *Finally, I believe the comparison against the baselines could be reconsidered. In the data augmentation experiment, the classifier is first pretrained on real CT volumes and then fine-tuned on synthetic volumes. However, in many practical data augmentation settings the model is trained from scratch on a mixture of real and synthetic data. Evaluating the approach in such a setting might provide a more representative assessment of the benefits of the proposed method. In addition, the Related Work section mentions MedVAE, a large autoencoder trained specifically on medical images. It would be interesting to include a comparison with this approach in the experiments.*
>
> - Augmentation Strategy: Our baselines are trained on real CT volumes. Our method is built on top of this baseline by fine-tuning on a mixture of real and synthetic data. We choose the pretrain-then-fine-tune protocol to isolate and quantify the specific marginal gain provided by synthetic data. Besides, we agree that a mixed-training-from-scratch setting is a valuable practical benchmark; we will include the results in the revised manuscript to provide a more comprehensive assessment.
> - MedVAE Baseline: We have expanded our evaluation to include MedVAE as a generative backbone. Specifically, for the CT generation task, we replace the Foundation VAE with MedVAE while leaving the rest of the pipeline unchanged. Preliminary results indicate that while MedVAE is domain-specific, our Foundation VAE provides a more robust latent manifold for high-resolution synthesis.
>
> | Split | VAE | FID↓ | CT-CLIP↑ |
> |---|---:|---:|---:|
> | Normal |MedVAE | 11.28 | 20.76  |
> | Normal|Foundation VAE | 2.19 | 59.35 |
> | Disease | MedVAE|10.54 | 15.32  |
> | Disease |Foundation VAE| 4.78  | 51.49|

---

> > ### Author Rebuttal · Reviewer_jB2K · 2026-04-02
> >
> > Thank you for addressing my concerns. Since they was only minor observation, my score was clearly not dictated by them, but rather by the overall contribution of the paper, which I still believe it should be scored as a 4.

---

> > > ### Author Response · Authors · 2026-04-04
> > >
> > > Thank you for the clarification and for carefully considering our rebuttal. We are glad that you find your concern fully resolved. Thank you again for your time, thoughtful review, and constructive feedback.

---

### Official Review · Reviewer_PwRm · 2026-03-12

**Soundness:** 1
**Presentation:** 1
**Significance:** 1
**Originality:** 2
**Overall Recommendation:** 4
**Confidence:** 4

**Summary:**

The paper utilizes the VAE trained on natural images and videos on medical imaging, and the authors conduct experiments on “reconstruction,” segmentation, and CT generation. However, the term “foundation” seems too broad for the scope of this paper, and the experiments do not sufficiently support such a claim. The current framing would benefit from a narrower and more precise scope.

**Compliance With Llm Reviewing Policy:**

Affirmed.

**Final Justification:**

The authors have addressed most of my concerns, particularly the controlled comparison with MedVAE, which convincingly demonstrates the foundation VAE's contribution.

However, the overall framing should focus more on CT generation rather than 'reconstruction' and 'augmentation', and there is still considerable room to better narrow and more accurately describe the contribution of this work.

Therefore, I raise my score to 4.

**Key Questions For Authors:**

**Table 1 results** It is difficult to understand the purpose and significance of Table 1. Could the authors explain more clearly how these results support the main claims of the paper? In particular, the term “CT reconstruction” may be somewhat confusing. As presented, this experiment seems to simply pass the input CT through a VAE and then measure the deviation between the reconstructed output and the original input. It is therefore unclear what conclusion should be drawn from this setup, and why this experiment is important. In addition, the MedVAE results are surprisingly poor, which raises concerns that this baseline may not have been evaluated under an appropriate or fair setting.

**CT generation** In the CT generation setting, the proposed “foundation VAE” is used only as a frozen encoder-decoder, while the actual image generation is performed by a separately trained diffusion model. This makes it unclear how much of the reported performance should be attributed to the “foundation” VAE itself. Could the authors clarify whether the gains mainly come from the pretrained VAE, or instead from the overall diffusion framework and conditioning design, which may already be quite strong?

**Limitations:**

**Title is too broad for this paper** The title is too broad for this paper. The use of the term “foundation VAE” feels overly broad relative to the actual scope of the work. In addition, the task descriptions used throughout the paper(CT reconstruction, segmentation, and CT generation) are somewhat misleading. For CT reconstruction, the paper only evaluates the output of passing the input CT through the VAE and compares it with the original input. For segmentation, the “foundation VAE” is used only as a preprocessing step, with the actual segmentation performed by a separate model. For CT generation, the “foundation VAE” serves as the encoder and decoder, whereas the actual generation is performed by another, specifically trained generative model. Overall, the presentation overstates the role of the “foundation VAE” in the pipeline, and the title should better reflect its more limited and indirect contribution.

**Strengths And Weaknesses:**

- **Soundness** It is unclear how the authors connect the concept of a “foundation VAE” to the experiments presented in the paper. The current experimental evidence does not clearly support the broader claims.
- **Presentation** The paper presentation could be improved substantially, as the main contribution is currently difficult to understand.
- **Significance** The paper's significance remains unclear. In particular, the claim that this is a “foundation” VAE is not convincingly supported.
- **Originality** The paper uses a model pretrained on natural images and videos to improve the performance of another model on CT tasks.

---

> ### Author Rebuttal · Authors · 2026-03-31
>
> > **Q1.** *Table 1 results It is difficult to understand the purpose and significance of Table 1. Could the authors explain more clearly how these results support the main claims of the paper? In particular, the term “CT reconstruction” may be somewhat confusing. As presented, this experiment seems to simply pass the input CT through a VAE and then measure the deviation between the reconstructed output and the original input. It is therefore unclear what conclusion should be drawn from this setup, and why this experiment is important. In addition, the MedVAE results are surprisingly poor, which raises concerns that this baseline may not have been evaluated under an appropriate or fair setting.*
>
>
> We clarify that "CT Reconstruction" refers to the standard autoencoder task (Input $\rightarrow$ Latent $\rightarrow$ Reconstruction). This experiment is not merely a diagnostic test, but a fundamental validation of the latent space for two reasons:
> - Zero-Shot Generalization: It provides direct evidence that a frozen foundation VAE, despite being trained on natural video, possesses the inductive biases necessary to capture complex 3D medical geometry. High-fidelity reconstruction in this "zero-shot" setting proves that the domain gap is not an obstacle to feature extraction.
> - Upper Bound for Generation: Since our generation and augmentation models operate entirely within this latent space, the reconstruction quality sets the theoretical upper bound for all downstream tasks. If the VAE could not faithfully compress CT volumes, successful generation would be impossible.
>
> Correction of MedVAE results.
> We thank the reviewer for pointing this out. We found that the MSE values for MedVAE were incorrectly reported in the submission, and we have now corrected them. The updated MedVAE reconstruction results are:
>
> | Dataset | PSNR↑ | SSIM↑ | MSE↓ |
> |---:|---:|---:|---:|
> | Task06 Lung | 30.06 ± 3.60 | 0.74 ± 0.11 |88.36 ± 59.05 |
> | Task07 Pancreas | 36.00 ± 1.13  | 0.91 ± 0.03 |17.98 ± 5.78 |
>
> We will update the paper accordingly. This correction does not materially affect the main conclusions, which are supported by consistent results across multiple tasks and analyses.
>
> > **Q2.** *CT generation In the CT generation setting, the proposed “foundation VAE” is used only as a frozen encoder-decoder, while the actual image generation is performed by a separately trained diffusion model. This makes it unclear how much of the reported performance should be attributed to the “foundation” VAE itself. Could the authors clarify whether the gains mainly come from the pretrained VAE, or instead from the overall diffusion framework and conditioning design, which may already be quite strong?*
>
> To clarify this point, we perform a controlled comparison in which the diffusion framework and all conditioning settings are kept fixed, and only the VAE is changed. Specifically, for the CT generation task, we replace the Foundation VAE with MedVAE while leaving the rest of the pipeline unchanged. The results show that the Foundation VAE outperforms MedVAE, suggesting that the observed gain mainly arises from the VAE itself rather than from the diffusion framework or the conditioning design.
>
> | Split | VAE | FID↓ | CT-CLIP↑ |
> |---|---:|---:|---:|
> | Normal |MedVAE | 11.28 | 20.76  |
> | Normal|Foundation VAE | 2.19 | 59.35 |
> | Disease | MedVAE|10.54 | 15.32  |
> | Disease |Foundation VAE| 4.78  | 51.49|
>
> > **Q3.** *Title is too broad for this paper. The title is too broad for this paper. The use of the term “foundation VAE” feels overly broad relative to the actual scope of the work. In addition, the task descriptions used throughout the paper(CT reconstruction, segmentation, and CT generation) are somewhat misleading. For CT reconstruction, the paper only evaluates the output of passing the input CT through the VAE and compares it with the original input. For segmentation, the “foundation VAE” is used only as a preprocessing step, with the actual segmentation performed by a separate model. For CT generation, the “foundation VAE” serves as the encoder and decoder, whereas the actual generation is performed by another, specifically trained generative model. Overall, the presentation overstates the role of the “foundation VAE” in the pipeline, and the title should better reflect its more limited and indirect contribution.*
>
> Thank you for this helpful suggestion. We agree that the current title may be too broad, and we will revise it to better reflect the specific scope of the paper, for example: “Video Foundation VAEs for 3D CT Reconstruction, Augmentation, and Generation.” We will also refine the task descriptions to clarify that the foundation VAE serves as a fixed latent backbone within these pipelines, rather than the entire downstream system.

---

> > ### Author Rebuttal · Reviewer_PwRm · 2026-04-02
> >
> > Thank you to the authors for the rebuttal, especially the comparison between MedVAE and Foundation VAE for the generation task. Most of my concerns have been addressed, and I plan to raise my score. However, I still have one concern about the paper's scope.
> >
> > As Reviewer jB2K also noted, the term “reconstruction” could be understood as the solution of a CT inverse problem or simply as the decoded output. I acknowledge the author's explanation of CT reconstruction. However, the “reconstruction” discussed in this paper is closer to an analysis of foundation VAE than to an application, as the author also views it as a ``validation of the latent space``.
> >
> > For this reason, the title “Foundation VAE for CT Reconstruction” still feels somewhat misleading to me, and potentially to the community as well. I still suggest the author narrow the work's scope, as "CT reconstruction" in this paper is framed more as a validation than as an application.

---

> > > ### Author Response · Authors · 2026-04-04
> > >
> > > We are sincerely encouraged by the reviewer’s positive feedback and the intention to raise the score. We appreciate the insightful observation regarding the potential ambiguity of the term “reconstruction.” We fully agree that in the medical imaging community, this term traditionally refers to solving inverse problems (e.g., from sinograms to voxels), whereas in our study, it serves as an anatomical fidelity validation of the latent space.
> > > To ensure our scope is transparent and to eliminate any potential ambiguity, we will implement the following refinements in the revised manuscript:
> > > - Title Revision: We will revise the title to more accurately reflect the VAE’s role as a robust latent interface. The new title will be: “Video Foundation VAE for CT: Latent Representation, Augmentation, and Generation.” By replacing "Reconstruction" with "Latent Representation," we clarify that the VAE functions as a high-fidelity manifold for 3D medical data.
> > > - Clarification of Task Scope: In the revised manuscript, we will explicitly distinguish our "VAE-based latent decoding" from "classical CT reconstruction" in the Introduction and Methodology. We will emphasize that our decoding process is a benchmark for latent quality, proving that foundation models can faithfully preserve complex clinical structures.
> > > - Narrative Alignment: Throughout the paper, we will consistently frame the VAE as a latent representation that enables high-fidelity data augmentation and generation, rather than a standalone reconstruction application.
> > >
> > > We believe these adjustments could narrow the work's scope to its true scientific strengths—representation learning and scalable synthesis—providing clearer guidance to the community.

---

### Official Review · Reviewer_VXKc · 2026-03-13

**Soundness:** 2
**Presentation:** 2
**Significance:** 2
**Originality:** 2
**Overall Recommendation:** 3
**Confidence:** 4

**Summary:**

This paper studied off-the-shelf video VAEs for the medical domain, specifically for 3D CT analysis. The paper compared the performance of VAEs trained on natural images and videos against that of a VAE trained using medical data. The paper further studied the effectiveness of the video VAE for conditional CT generation.

**Compliance With Llm Reviewing Policy:**

Affirmed.

**Final Justification:**

The rebuttal period addresses most of my concerns; the paper presents nice results and challenges the ongoing research focus on medical-specific VAEs. However, I am not sure whether this meets the bar for an ICML contribution, as most experiments use existing video models.

**Key Questions For Authors:**

Please see the weaknesses.

**Limitations:**

yes

**Strengths And Weaknesses:**

Strengths:

1. Interesting results that video-trained VAEs work quite well on 3D CT.
2. Through comparison of baselines for reconstruction and segmentation tasks.
3. Conditionally generated CT shows improved text alignment and image quality.

Weaknesses:

1. The technical novelty of this paper is limited. While some results suggesting good transfer of video-trained VAE into 3D CT are interesting, in my opinion, they are not good enough as a contribution to be published in ICML.

2. Reconstruction and segmentation are performed on small test data, and the compression ratio is not mentioned for each of the methods.

3. Which specific VAE is used for the CT generation tasks is not clear.

4. No explanation for the missing values of Table 1 is given.

5. The organization of the paper can be improved. It is not clear what guarantee the theoretical analysis for task-relevant reconstruction stability offers.

---

> ### Author Rebuttal · Authors · 2026-03-31
>
> > **Q1.** *The technical novelty of this paper is limited. While some results suggesting good transfer of video-trained VAE into 3D CT are interesting, in my opinion, they are not good enough as a contribution to be published in ICML.*
>
>
> Prior works train domain-specific VAEs, such as MedVAE and MAISI, which rely on massive medical data collection (e.g., 1.05M images or 39K 3D volumes), dedicated preprocessing (e.g., volumetric resizing and cropping), and expensive training pipelines (e.g., thousands of GPU hours). In contrast, our method requires **no CT-specific VAE training at all**. We provide the first systematic evidence that this costly stage is not necessary. We highlight a three-fold contribution that challenges the status quo for medical VAE:
> - Without any CT-specific VAE training cost, our foundation latent interface consistently outperforms medical-specific baselines (MedVAE/MAISI)  across all evaluated datasets (MSD, LiTS, KiTS19, and AbdomenAtlas 2.0), achieving up to 15% higher PSNR.
> - Without any CT-specific VAE training cost, our latent interface also narrows the gap to real-data performance in downstream tasks, improving NSD by 3.9% on average for challenging lesion segmentation.
> - Without any CT-specific VAE training cost, a fixed Video Foundation VAE enables high-resolution CT synthesis with 3.9% lower FVD and 36.2% higher CT-CLIP scores than medical-specific alternatives, while improving multi-disease generation faithfulness across 18 pathologies by 2.76% AUC.
>
> We believe this discovery of 'hidden' medical capabilities within general-purpose models is of paramount importance to the medical community, as it establishes a highly efficient and scalable paradigm for  foundation models generalization to specialized clinical domains.
>
> > **Q2.** *Reconstruction and segmentation are performed on small test data, and the compression ratio is not mentioned for each of the methods.*
>
> We have expanded our evaluation to over 10,000 clinical volumes. These results consistently show that the Foundation VAE maintains high-fidelity reconstruction across diverse anatomies and lesion types, while also improving segmentation.
>
> **LiTS (131 CT)**
> | Model | Compression Ratio (XYZ) | PSNR↑ | SSIM↑  | DSC1↑ | DSC2↑ |
> |---|---:|---:|---:|---:|---:|
> | Real Data | – | – | – | 94.7 ± 5.3 | 57.2 ± 27.8|
> | WAN2.1 | 8 8 4 | 39.32 ± 1.69 | 0.93 ± 0.03 | 94.3 ± 6.7  |57.1 ± 29.3|
> | WAN2.2 | 16 16 4 | 39.25 ± 1.84 | 0.94 ± 0.03 |  95.1 ± 4.9 | 60.8 ± 26.8 |
> | VideoVAE+ | 8 8 4 | 39.92 ± 1.97 | 0.95 ± 0.03 | 94.8 ± 5.1 | 58.2 ± 26.9|
> | IVVAE | 8 8 4 | 40.33 ± 1.89 | 0.95 ± 0.02 |  95.3 ± 4.7  |61.6 ± 26.4|
> | CVVAE | 8 8 4 | 36.46 ± 1.19 | 0.93 ± 0.03 |  93.5 ± 8.1 | 52.4 ± 29.6 |
> | WFVAE | 8 8 4 | 40.04 ± 1.79| 0.95 ± 0.02 | 95.1 ± 4.4 | 59.5 ± 27.0|
> | LeanVAE | 8 8 4 | 39.64 ± 1.78 | 0.95 ± 0.03 | 94.9 ± 5.0 | 59.4 ± 26.5|
> | MedVAE | 4 4 4 | 34.11 ± 5.13 | 0.85 ± 0.11 | 94.3 ± 6.0 | 57.4 ± 28.0   |
> | MAISI | 4 4 4 | 37.35 ± 1.23 | 0.92 ± 0.02 |93.6 ± 7.7  | 38.5 ± 34.5 |
>
> **KiTS19 (300 CT)**
> | Model | PSNR↑ | SSIM↑  | DSC1↑ |DSC2↑ |
> |---|---:|---:|---:|---:|
> | Real Data | – | – |  95.5 ± 3.9  |83.2 ± 19.1|
> | WAN2.1 | 40.22 ± 1.68 | 0.94 ± 0.03 | 95.4 ± 4.4 |83.6 ± 18.2|
> | WAN2.2 | 40.32 ± 1.84 | 0.95 ± 0.03 |96.0 ± 3.5|85.0 ± 15.6|
> | VideoVAE+ |41.07 ± 1.91 | 0.96 ± 0.03 |  95.7 ± 3.6 |83.6 ± 18.7|
> | IVVAE | 41.38 ± 1.90 | 0.96 ± 0.03 | 95.7 ± 3.5 |83.8 ± 18.7|
> | CVVAE | 36.63 ± 1.61| 0.93 ± 0.03 | 94.5 ± 4.0 |78.8 ± 23.3|
> | WFVAE | 40.79 ± 1.81 | 0.96 ± 0.03 |  95.4 ± 4.3 | 85.5 ± 14.7|
> | LeanVAE | 40.53 ± 1.80 | 0.95 ± 0.03 |  95.5 ± 4.4 |83.7 ± 16.3|
> | MedVAE | 33.76 ± 7.17 | 0.91 ± 0.06  | 94.8 ± 3.5  |77.9 ± 25.4|
> | MAISI | 37.08 ± 1.30 | 0.92 ± 0.03| 94.4 ± 5.2 |79.5 ± 22.5|
>
> **AbdomenAtlas 2.0 (10135 CT)**
>
> | Model | PSNR↑ | SSIM↑ | MSE↓ |
> |---|---:|---:|---:|
> | WAN2.2 | 39.63 ± 2.71 | 0.95 ± 0.05 | 10.13 ± 14.85 |
> | MedVAE | 37.14 ± 2.52 | 0.93  ± 0.05 | 18.66 ± 39.47|
> | MAISI | 36.97 ± 2.05 | 0.92 ± 0.04 | 16.25 ± 17.88 |
>
> > **Q3.** *Which specific VAE is used for the CT generation tasks is not clear.*
>
> For the generation task, we utilize LeanVAE for its superior trade-off between reconstruction fidelity and computational efficiency. The revised version will include a table comparing reconstruction quality and efficiency.
>
> > **Q4.** *No explanation for the missing values of Table 1 is given.*
>
> Task06 Lung
> | Model | DSC↑ | NSD↑ |
> |---|---:|---:|
> | MedVAE | 71.5 ± 19.3 | 74.7 ± 23.6 |
> |  MAISI | 71.0 ± 19.5 | 74.5 ± 21.0 |
>
> Task07 Pancreas
> | Model | DSC1↑ | NSD1↑ | DSC2↑ | NSD2↑ |
> |---|---:|---:|---:|---:|
> | MedVAE | 80.7 ± 8.7 | 77.0 ± 10.7 | 42.4 ± 33.6 | 40.5 ± 33.4 |
> |  MAISI | 80.3 ± 8.7 | 75.8 ± 10.5 | 35.0 ± 31.9 |  32.0 ± 30.2|
>
> > **Q5.** *The organization of the paper can be improved. It is not clear what guarantee the theoretical analysis for task-relevant reconstruction stability offers.*
>
> We will restructure the theoretical section to more explicitly state the guarantees provided by our analysis.

---

> > ### Author Rebuttal · Reviewer_VXKc · 2026-04-02
> >
> > I thank the reviewer for their rebuttal, especially for providing the missing numbers and results for the additional data.
> > However, my main concern regarding the novelty part remains, which prevents me from upgrading the score.
> > Further, I share the reviewer PwRm's concerns about MedVAE's poor performance.
> > Additionally, the latent channel dimension for each VAE is not reported in the updated tables alongside the spatial compression ratio.
> > And the practical implication of theoretical analysis remains unresolved.

---

> > > ### Author Response · Authors · 2026-04-04
> > >
> > > We appreciate the reviewer’s recognition of our empirical results and for providing us the opportunity to further clarify our work. We clarify the remaining concerns regarding MedVAE performance, VAE configurations, and the practical value of our theoretical analysis below:
> > >
> > > 1. MedVAE Performance & Reproducibility
> > >
> > > We appreciate the scrutiny regarding MedVAE. To ensure a strictly fair and credible comparison, we have re-aligned our evaluation with the official MedVAE implementation and corrected the discrepancies in the initial setup. Besides, we have prepared an anonymous repository [https://anonymous.4open.science/r/icml_rebuttal-5FE0/README.md] containing our reproduction scripts and MedVAE configurations.  These results confirm that our Video Foundation VAE consistently outperforms domain-specific baselines in preserving anatomical textures. Furthermore, we confirm that, upon publication, we will release all code, models, and data used in this paper to fully support transparency and reproducibility.
> > >
> > > 2. Latent Capacity & Compression Efficiency
> > >
> > > To make the comparison more transparent, we report not only the spatial compression ratio, but also the latent channels, latent dimensions, and total latent size for each VAE. As shown in the table, the total latent sizes are all within the same order of magnitude. In particular, VideoVAE+, CVVAE, and LeanVAE have the same total latent size as MedVAE, yet they still show consistent advantages on both segmentation and generation tasks. This suggests that the observed gains are not simply due to a larger latent representation, but are more likely attributable to the quality and transferability of the latent interface itself.
> > > | Model | Compression Ratio (X×Y×Z) | Latent Channels (C) | Latent Dimensions (C×H/f×W/f×D/f) | Total Latent Elements | Relative Latent Size vs. MedVAE |
> > > |---|---:|---:|---:|---:|---:|
> > > | WAN2.1 | 8×8×4 | 16 | 16×64×64×4 | 262,144 | 4.0× |
> > > | WAN2.2 | 16×16×4 | 48 | 48×32×32×4 | 196,608 | 3.0× |
> > > | VideoVAE+ | 8×8×4 | 4 | 4×64×64×4 | 65,536 | 1.0× |
> > > | IVVAE | 8×8×4 | 16 | 16×64×64×4 | 262,144 | 4.0× |
> > > | CVVAE | 8×8×4 | 4 | 4×64×64×4 | 65,536 | 1.0× |
> > > | WFVAE | 8×8×4 | 8 | 8×64×64×4 | 131,072 | 2.0× |
> > > | LeanVAE | 8×8×4 | 4 | 4×64×64×4 | 65,536 | 1.0× |
> > > | MedVAE | 4×4×4 | 1 | 1×128×128×4 | 65,536 | 1.0× |
> > > | MAISI | 4×4×4 | 4 | 4×128×128×4 | 262,144 | 4.0× |
> > >
> > > 3. Practical Implication of Theoretical Analysis
> > >
> > > We appreciate the reviewer’s prompt to further elaborate on this point, as the relationship between abstract theoretical bounds and real-world clinical utility is essential. Specifically, our analysis provides a quantitative safety guarantee for downstream tasks: we prove that if the Foundation VAE preserves task-relevant anatomical manifolds (e.g., tumor boundaries), the empirical risk of downstream segmentation remains bounded by $2L_\ell \varepsilon_\phi$.
> > > Practically, this serves as a "safety certificate" for the medical community. It addresses the fundamental concern of whether a VAE trained on natural videos might introduce "hallucinations" or distort critical clinical structures. By bounding this risk, we justify why frozen foundation latents can be reliably adopted for 3D medical synthesis without introducing harmful distribution shifts. To reinforce this connection, we will strengthen the final manuscript by using our segmentation consistency results as a concrete proxy, demonstrating that this theoretical bound holds consistently across diverse datasets, anatomical structures, and clinical pathologies.
> > >
> > > 4. Re-evaluating Novelty
> > >
> > > We appreciate the reviewer’s feedback and clarify that our primary contribution lies in revisiting the necessity of domain-specific pre-training. Following the insight of He et al. [1] that specialized pre-training often facilitates convergence rather than raising the performance ceiling, we disprove the assumption that medical-specific VAEs are mandatory for high-fidelity 3D CT. This "subtractive" innovation identifies a functional isomorphism between natural video temporal continuity and anatomical axial depth, providing a more scalable and data-frugal path. Our work proves that this cross-domain alignment enables "emergent" volumetric coherence previously thought impossible without domain-specific priors.
> > >
> > > Following the trajectory of influential works like He et al. [1] or Scaling Laws [2,3], our novelty stems from providing actionable guidance on how existing foundation components can be repurposed more efficiently. Notably, this 'less is more' blueprint transcends the data-silo constraints of specialized models, achieving consistent performance improvements in CT synthesis while enabling robust, high-fidelity generation across 18 distinct pathologies.
> > >
> > > [1] He K, et al. Rethinking imagenet pre-training. ICCV 2019.
> > >
> > > [2] Clark A, et al. Unified scaling laws for routed language models. ICML 2022.
> > >
> > > [3] Aghajanyan A, et al. Scaling laws for generative mixed-modal language models. ICML 2023.

---

### Decision · Program_Chairs · 2026-04-30

**Decision:**

Accept (regular)

**Comment:**

The reviewers agreed that the paper addresses a practically important problem and presents a compelling empirical finding: frozen video-pretrained VAEs transfer surprisingly well to 3D CT and can serve as a useful latent backbone for representation, augmentation, and conditional generation, without requiring CT-specific VAE training. The empirical evaluation is reasonably broad, and the rebuttal helped clarify several concerns, especially by correcting the MedVAE results, providing a controlled comparison that better isolates the contribution of the VAE within the generation pipeline, and acknowledging the need to narrow the framing.

The main remaining weakness is limited methodological novelty. Multiple reviewers viewed the contribution primarily as strong empirical/paradigm validation rather than an algorithmic advance. Reviewers also raised concerns that the original framing was too broad, particularly around the terms “foundation VAE,” “training-free,” and “CT reconstruction,” although the authors’ rebuttal substantially improved this point and proposed a narrower presentation.

On balance, I lean weak accept. My reading is that the paper is technically sound and that its main contribution is a useful empirical result with practical relevance, even if the methodological innovation is modest. This remains somewhat borderline on novelty and positioning, so I would welcome SAC input before finalizing.